# Hedgehog-stimulated phosphorylation at multiple sites activates Ci by altering Ci–Ci interfaces without full Suppressor of Fused dissociation

Hoyon Kim, Jamie C. Little[¤a], Jiashen Li[¤b], Bryna Patel, Daniel Kalderon *

Department of Biological Sciences, Columbia University, New York, New York, United States of America

¤a Current address: Department of Molecular Life Sciences, University of Zurich, Zurich, Switzerland,
¤b Current address: Department of Molecular Genetics, Ohio State University, Columbus, Ohio, United States of America
* ddk1@columbia.edu

## Abstract

Hedgehog (Hh) proteins elicit dose-dependent transcriptional responses by binding Patched receptors to activate transmembrane Smoothened (Smo) proteins. Activated Smo inhibits Ci/Gli transcription factor phosphorylation by Protein Kinase A and consequent proteolytic processing to repressor forms; it also promotes nuclear transport and activity of full-length Ci/Gli proteins to induce Hh target genes. Smo-activated Fused (Fu) kinase drives Ci activation in *Drosophila*, while Suppressor of Fused (Su(fu)) counters full-length Ci/Gli activity and stabilizes full-length Ci/Gli by direct binding to at least three surfaces. Here, we used CRISPR-generated designer *ci* alleles to investigate alterations to Fu phosphorylation sites and to regions around Ci–Su(fu) interfaces under physiological conditions in *Drosophila* imaginal wing discs. Surprisingly, we identified alterations that activate Ci without significant loss of stabilization by Su(fu) and contributions of multiple Fu target sites to Ci activation in the absence of Su(fu), suggesting that the affected sites mediate Ci activation by regulating Ci–Ci, rather than Ci–Su(fu) interactions. We propose that those interactions maintain full-length Ci in a closed conformation that also facilitates, and is stabilized by, cooperative Ci–Su(fu) binding. Access to binding partners necessary for Ci activation is promoted through phosphorylation of at least four Fu sites on Ci, likely by directly disrupting Ci–Ci contacts and one Ci–Su(fu) interface without substantial Ci–Su(fu) dissociation, contrary to previous proposals. We also found that the Ci binding partner, Costal 2 (Cos2), which silences Ci in the absence of Hh, can facilitate Ci activation by Fu kinase.

## Introduction

Extracellular Hedgehog (Hh) signaling proteins constitute a major means of cell communication in vertebrates and in *Drosophila*, where they were discovered [1–5].

**Data availability statement:** All relevant data are within the paper and its Supporting Information files.

**Funding:** Funding was provided by the National Institutes of Health (NIH RO1GM41815 and NIH RO3HD111663 awarded to DK). The funders had no role in study design, data collection and analysis, decision to publish, or preparation of the manuscript.

**Competing interests:** The authors have declared that no competing interests exist.

**Abbreviations:** Ci, Cubitus interruptus; CK1, Casein Kinase 1;Cos2, Costal-2;Fu, Fused;gRNA3, guide RNA 3;GSK3, Glycogen Synthase Kinase 3;Hh, Hedgehog;PKA, Protein Kinase A; Smo, Smoothened;Su(fu), Drosophila suppressor of fused; Sufu, Vertebrate or generic (Drosophila or vertebrate) suppressor of fused.

Numerous human developmental defects and syndromes result from genetic alterations leading to either loss or gain of Hh signaling and are closely mirrored in mice [6,7]. Activating Hh pathway mutations are also associated with many types of cancer, especially basal cell carcinoma and medulloblastoma [8–12]. Hence, greater understanding of Hh signal transduction has a potentially large benefit for forecasting, diagnosing, and combating several common, major human diseases and disorders.

An outline of Hh signal transduction emerged from early *Drosophila* genetic studies and major elements proved to be conserved in vertebrates [1,2,13]. The first steps universally involve the 12-transmembrane-domain Patched (Ptc) family of receptors and the seven-transmembrane-domain Smoothened (Smo) family, which shares some characteristics with G-protein coupled receptors. Ptc constitutively keeps Smo inactive, but this can be reversed by Hh binding to Ptc. Activated Smo is, in all cases, the initiator of cytoplasmic signal transduction.

The "canonical" result of signaling is alteration of the activity of Gli-family zinc-finger DNA-binding transcription factors known as Cubitus interrupts (Ci) in *Drosophila* or Glioma-associated oncogene (Gli) proteins in vertebrates. In the absence of active Smo, Ci, Gli2, and Gli3 full-length proteins are proteolytically processed from the C-terminus to form truncated transcriptional repressors, which accumulate in the nucleus [14,15]. The steps preceding proteolytic processing are sufficiently slow that the full-length proteins are present at significant steady-state levels, generally similar to processed derivative levels. They are, however, held inactive, largely in the cytoplasm, when there is no Hh signal. Activated Smo inhibits proteolytic processing in proportion to Hh dose [16] to reduce the concentration of Ci/Gli repressors and preserve more full-length protein for potential activation. Crucially, active Smo also converts the full-length Ci/Gli proteins to effective transcriptional activators ("activation"), including slightly enhancing nuclear accumulation [17,18].

Through the dual responses of inhibiting Ci/Gli processing and promoting Ci/Gli activation, various genes with Gli/Ci binding sites in their regulatory regions are de-repressed and activated by Hh. Most "Hh target genes" are tissue specific, but *ptc* genes are near universal, dose-dependent targets and are therefore excellent reporters of Hh pathway activity. The Ptc circuitry likely reflects an ancient evolutionary origin of the pathway and, in the context of multicellular eukaryotes, leads to a more rapid decline in the spatial profile of Hh ligands over a field of responding cells because binding to Ptc leads to Hh internalization and degradation [19,20]. In several settings, including patterning of the mouse neural tube and *Drosophila* wing imaginal discs, Hh signals over a range of several cells and act as a morphogen, with functionally significant, distinct outcomes in cells receiving different doses of Hh ligand [2,3].

The mechanism of proteolytic processing is quite well understood and conserved. Protein Kinase A (PKA) phosphorylates a cluster of sites in Ci/Gli, leading to successive priming of several additional local phosphorylation events, involving Casein Kinase 1 (CK1) and Glycogen Synthase Kinase 3 (GSK3), to create a binding site for the Slimb/β-TRCP recognition subunit of a Cul1 E3 ligase [21–23]. Ubiquitinated

Ci/Gli then associates with the 26S proteasome, but progressive digestion from the C-terminus is arrested to spare a truncated transcriptional repressor. The initial phosphorylation of Ci/Gli proteins by PKA, CK1, and GSK3 is facilitated by scaffolding functions of kinesin-family proteins, Costal-2 (Cos2) in *Drosophila* or Kif7 in vertebrates [2,24]. Cos2 binds directly to at least three regions of Ci [25] and to Fu, which likely contributes directly to the scaffolding function that promotes processing [26,27]. The contribution of Fu to Ci processing in the absence of Hh does not require Fu kinase activity.

In *Drosophila*, activation of Smo involves increased protein stability, conformational changes and extensive Smo phosphorylation, with the net result that more Cos2/Fu complex binds to Smo, and likely in a different manner [28–31]. The exposure of a pseudo-substrate site for PKA on activated Smo, supported by additional PKA–Smo interactions, then attenuates PKA activity towards Ci/Gli proteins, as observed also in vertebrates [26,32,33]. In *Drosophila*, some results suggest that Smo activation also leads to partial dissociation of PKA, CK1, or GSK3 from Cos2 and partial dissociation of Ci from Cos2 [24,34].

The mechanisms regulating Ci/Gli activity are less well understood but have emphasized the role of Suppressor of fused ("Sufu" in reference to vertebrates or all species collectively; "Su(fu)" for *Drosophila*). Sufu is a conserved direct Ci/Gli binding partner that can limit full-length Ci/Gli activity [18,35]. In *Drosophila*, the inhibitory effect of Sufu is evident only in situations where Ci processing is blocked. This is likely because Cos2 suffices as a direct stoichiometric inhibitor provided full-length Ci does not accumulate in excess of Cos2 binding capacity [18]. In mice, loss of Sufu alone causes strong ectopic Hh pathway activation [36]. This difference likely reflects weaker association of Gli with Kif7 than for Ci/Cos2 and the fact that low levels of pathway activity elicit transcriptional activation of Gli-1 and consequent positive feedback [2,37,38]. It has also been reported that Sufu can, unlike in *Drosophila*, facilitate Gli processing in mice [39,40]. Despite these contextual differences and notably different Sufu mutant phenotypes, the biochemical mechanism by which Sufu limits full-length Ci/Gli activity is plausibly conserved [38]. Here, we explore that mechanism and how it is opposed by Hh in *Drosophila*.

Very little is known about the molecular connection between active Smo and activation of Gli proteins. In contrast, there is extensive evidence that the Fused (Fu) protein kinase is the key link in *Drosophila*. Fu is activated only when Smo is activated [41]. This process requires cross-phosphorylation of minimally active Fu molecules bound to Cos2 and presumably results from clustering consequent to changes in Smo–Cos2–Fu interactions when Smo is activated [42–44]. Hh target gene induction in *Drosophila* wing discs is severely compromised by kinase domain point mutations that abrogate Fu kinase activity (such as *fu*[mH63]), even though the regulation of Ci proteolytic processing is unaltered [18,27,45,46]. Conversely, clustering and cross-phosphorylation of Fu can be engineered synthetically in the absence of Smo activation, using a "GAP-Fu" or "Fu-EE" transgene, and can suffice for strong Ci activation [43,47]. GAP-Fu has the palmitoylated N-terminal domain of GAP-43 followed by Fu coding region to localize the fusion protein to the plasma membrane, facilitating clustering [47], while Fu-EE has acidic substituents at two residues in the activation loop normally phosphorylated during activation of Fu as a protein kinase [43]. Thus, both loss and gain of function experiments show that Fu kinase activity is the key mediator of Ci activation.

Multiple Fu-dependent, likely direct, phosphorylation sites were identified on Ci and its binding partners, first by SDS-PAGE mobility shifts and then by more systematic mass spectrometry assays conducted under synthetic circumstances of strong expression of activated Fu derivatives and potential substrates [38,43,48–50]. Interestingly, most of the identified sites can prime further local phosphorylation by CK1 (targeting S or T residues within a consensus of (S/T)pXX(X)(S/T)) and genetic evidence has shown that CK1 activity has a role in Ci activation by Fu [43,48]. The role of individual Fu sites has been analyzed by alterations to Ala or Val residues (often together with the following CK1 sites) or to acidic residues (together with the following CK1 sites) to mimic phosphorylation. Analysis of the functional role of these sites on Cos2 and Su(fu) has not been straightforward and has yielded some surprising results. Two sites (S572 and S931) were identified on Cos2 and were initially shown to affect Cos2–Ci association in vitro, Ci activation in some assays and Ci processing in others [49,50]. Each of these assays, including those performed in animals, used supra-physiological levels of altered

Cos2 proteins. Hh signaling depends critically on the relative stoichiometry of various binding partners and cannot, for example, be reconstituted faithfully in animals if either Cos2 or Ci is expressed in excess [16,27,43,51]. When Cos2 variants with S572A and S931A alterations, alone or combined, were tested at physiological levels, there was no change in Hh signaling [27]. When Su(fu) variants with either Alanine or acidic substituents of the identified Fu site and subsequently primed CK1 sites were tested, albeit in transgenes expressed at non-physiological levels, there were also no changes in several assays of Hh signaling [38,43]. There was also no change in Hh signaling when both the identified Cos2 and Su(fu) sites were altered to Ala residues, leading to the suggestion that there are likely key targets of Fu on Ci itself [27].

Several Fu-dependent sites on Ci were identified biochemically and their importance was tested initially by an assay in tissue culture using processing-resistant Ci molecules to select two clusters, initiated by Fu phosphorylation at S218 and S1230 as the most influential [48]. Genetic alterations of these two sites were then tested in *Drosophila* wing discs, but not using normal expression levels or patterns, to conclude that they were the primary sites for Ci activation by Fu [48]. Thereafter, an additional site (S1382) was uncovered and tested in the same way to conclude that it too was important [52]. Given the Cos2 precedent [27] and the known importance of testing Hh signaling component variants under physiological conditions, we explored the role of Ci phosphorylation site variants using CRISPR knock-in alleles [16] and testing responses to Hh and Fu kinase under a variety of genetic conditions, in order to understand better how Fu kinase activates full-length Ci. The results revealed the involvement of additional Fu sites and suggest a new model, in which Ci is activated principally by changes in Ci–Ci interactions rather than Ci–Su(fu) interfaces.

## Results

### Hh signaling in *Drosophila* wing discs; contributions of Su(fu) and Fu

Hh signal transduction has been productively studied through analysis of wing imaginal discs dissected from late third instar larvae. From specification in embryos until this stage, posterior wing disc cells heritably express En and do not mingle with anterior cells [53,54]. En promotes *hh* expression and represses *ci*, so that Hh protein made in posterior cells can signal only to anterior Ci-expressing cells [55]. This occurs in a stripe spanning 12–15 anterior cells adjacent to the posterior compartment, known as the AP (anterior-posterior) border [56]. Anterior cells beyond the AP border ("anterior cells") express Ci but do not receive significant Hh input. The following illustrated paragraphs explain key methods and prior observations for wild-type Ci that underlie current models of Hh signaling and provide the baseline controls for our investigation of Ci variant properties.

Within the AP border domain, there is Hh-dependent graded induction of *ptc*, commonly reported by a *ptc-lacZ* transgene. The posterior edge of *ptc-lacZ* marks the edge of the anterior compartment, with no detectable expression in posterior cells (Fig 1A). The most anterior cells of the AP border also have no detectable *ptc-lacZ* but express *dpp* and show some elevation of full-length Ci-155 protein (detected by the monoclonal 2A1 antibody, which does not recognize the proteolytically processed form, Ci-75) relative to anterior cells (Fig 1A'); loss of repression by processed Ci is sufficient for *dpp* but not *ptc-lacZ* expression [14,16,57]. En protein is induced transcriptionally by Hh in the 2–3 most posterior cells of the AP border, beginning only in third instar larvae [56,58]. Co-incidence with *ptc-lacZ* distinguishes Hh-induced anterior En from heritable En in posterior cells (Fig 1C). The full-length Ci profile does not increase uniformly toward the source of Hh; instead, it increases, peaks, and then declines over roughly the domain of anterior En expression (Fig 1A'–C') [16,18,59]. Although other contributions for this Hh-stimulated decline have been suggested, a substantial part of this decline has recently been attributed to reduced *ci* transcription in response to Hh-induced anterior En [60]. This mechanism would serve to moderate *ptc-lacZ* induction close to the Hh source. Hence, anterior En induction is the best measure of the highest levels of Hh signaling, with *ptc-lacZ* signal potentially saturating and relatively insensitive at the high end of the Hh signaling spectrum.

Genetic removal of Su(fu), using the *Su(fu)$^{LP}$* allele, which deletes a large segment of the coding region [61], greatly reduces Ci-155 levels in anterior cells and at the AP border to roughly the same extent (Fig 1D' and 1E') but does not

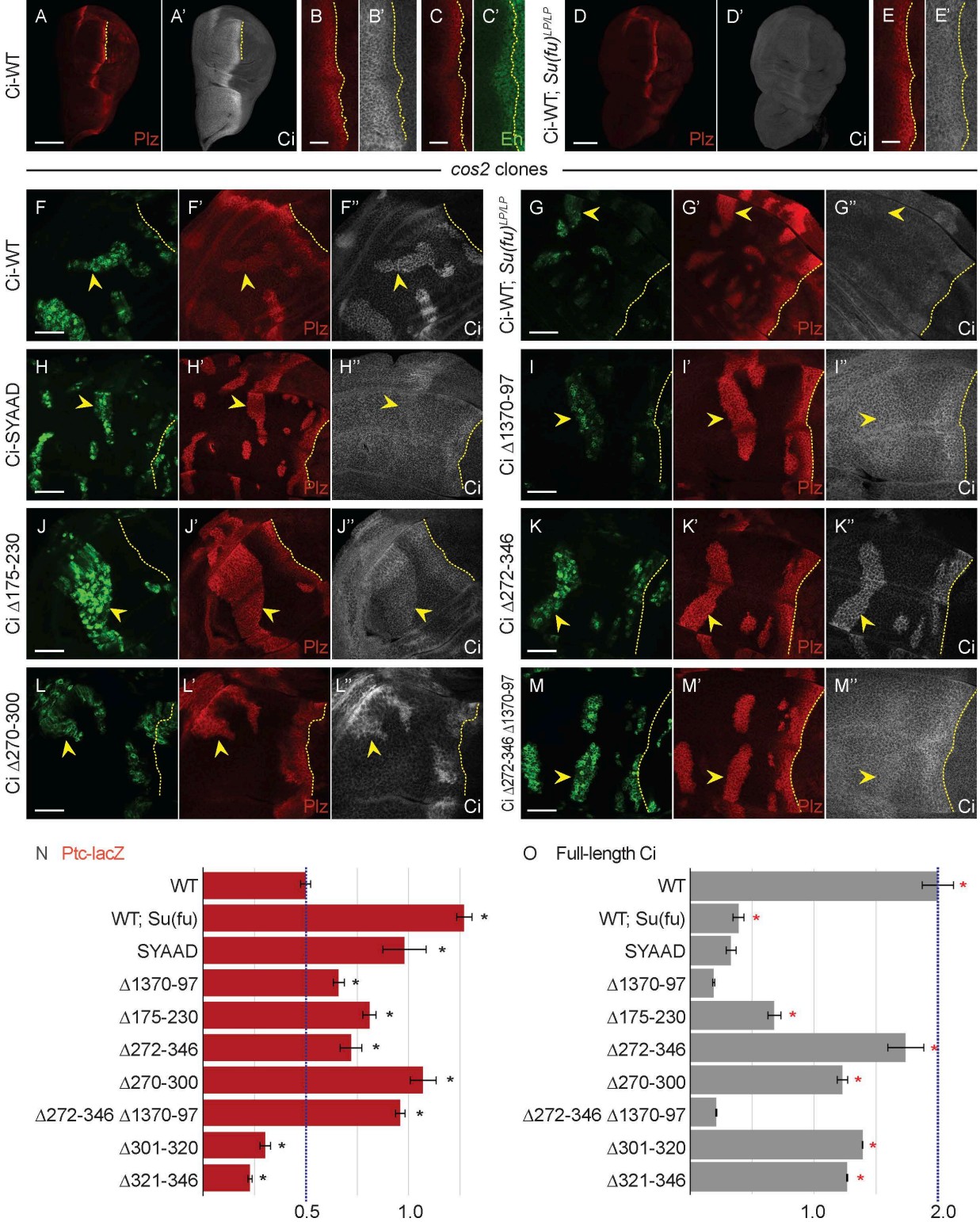

**Fig 1. Loss of Su(fu) binding sites or two adjacent regions from Ci increase its activity. (A–E)** Third instar wing discs with one copy of Ci-WT (*crCi-WT/ ci⁹⁴*), showing a stripe of induction of *ptc-lacZ* reporter expression ("Ptc-lacZ"), visualized by Beta-galactosidase antibody staining (red), in anterior cells near the center of the disc responding to Hh, released from posterior (right) cells. The posterior edge of Ptc-lacZ defines the AP

compartment boundary and is marked by yellow dotted lines. **(A', B', D, E")** Full-length Ci (Ci-155) visualized by 2A1 antibody staining (gray-scale), and **(C')** *en* expression visualized by En antibody staining (green). Anterior En induction by Hh is to the left of the AP border (yellow dotted lines) and co-incides with lower Ci-155 levels, which are otherwise elevated in Hh signaling territory. **(A–C')** shows wild-type discs using **(A–A')** 20× and **(B–C')** 63× objective lenses. **(D–E')** A *Su(fu)^{LP/LP}* disc using **(D–D')** 20× and **(E–E')** 63× objective lenses, showing normal Ptc-lacZ but lower levels of Ci-155 in anterior and AP border cells. **(F–M)** Third instar wing discs (63× objective) with one copy of the indicated *ci* CRISPR allele, GFP marking homozygous *cos2* mutant clones (green; yellow arrowheads), and yellow dotted lines marking the AP border. **(F'–M')** Ptc-lacZ expression (red) and **(F"–M")** Ci-155 expression (gray-scale) in the same discs. **(F')** Ptc-lacZ induction by Ci-WT in *cos2* mutant clones was **(G')** increased in discs that lack Su(fu) activity, and **(H'-M')** was increased to a lesser degree by each of the Ci variants shown. **(F")** Ci-155 levels are increased in *cos2* mutant clones because proteo-lytic processing is blocked, but the increase was reduced or absent **(G")** in the absence of Su(fu) and **(H"–M")** for most Ci variants, potentially due to reduced Su(fu)-dependent Ci stabilization or En-mediated inhibition of *ci* transcription; **(K")** Ci Δ272-346 and **(L")** Ci Δ270-300 have higher C-155 than other Ci variants. The brightness of **G" –J"** and **M"** images was increased relative to others in order to see any changes in Ci levels in clones more eas-ily. Quantitation used 20× images, all obtained under the same conditions. Scale bars are **(A, D)** 100 μm, **(B, C, E)** 20 μm, and (all others) 40 μm. **(N, O)** Bar graphs showing average **(N)** Ptc-lacZ intensity or **(O)** Ci-155 signal intensity in *cos2* clones relative to the AP border of control discs (with two copies of Ci-WT), together with SEMs (*n*-values 20, 10, 16, 22, 53, 29, 27, and 31, respectively, for each graph). Ci-WT ratios align with blue dotted lines. Differ-ences with $p < 0.005$ (Student *t* test with Welch correction) are indicated for comparing a Ci variant to **(N)** Ci-WT (black asterisk) or **(O)** Ci Δ1,370–1,397 (red asterisk). Please see Materials and methods for details of measurements and expression of all experimental values relative to AP border values of control wild-type wing discs, and S1 Data for raw data, including all *p* values. A cartoon of Ci domains and deletions, together with testing for En induc-tion in *cos2* clones, are shown in S1 Fig.

affect the expression of Hh target genes *ptc-lacZ* (Fig 1D and 1E) or En [18,43]. Ci-155 reduction is thought to result from increased protein degradation (distinct from proteolytic processing); an analogous response is seen for Gli proteins, but the mechanism is not known [18,62–64]. Loss of Fu kinase activity (*fu^{mH63}*) results in loss of anterior En and greatly reduced *ptc-lacZ* induction [18,43,45]. As expected from reduced induction of *ptc* transcription and hence Ptc protein, Hh spreads further [19], so that the weak *ptc-lacZ* stripe is broader than normal, as is the region of elevated Ci-155 due to inhibition of proteolytic processing (Fig 2A and 2B). High Ci-155 also extends right up to posterior cells (Fig 2A' and 2B') because Hh signaling is no longer strong enough, through En induction or other mechanisms, to stimulate its decline [16]. Wing discs with both *fu^{mH63}* and *Su(fu)^{LP}* mutations have normal levels and AP border width of Ptc-lacZ (Fig 2C), producing adults with normally patterned wings. Hence, it was deduced that Fu kinase activity is required to oppose the inhibition of Ci-155 activation by Su(fu) [65]. However, there is no anterior En expression in *fu^{mH63}; Su(fu)* mutant discs, leading to the deduction that Fu kinase must also activate Ci-155, directly or indirectly, in another way [43]. Alleles of *fu* that truncate or eliminate stable protein additionally impair Ci-155 processing; hence, *fu^{mH63}* is used to focus solely on the role of Fu kinase activity without accompanying changes in Ci-155 processing or Cos2-Fu and other Fu complexes [27,45,66,67].

The activity of Ci variants in the absence of activation by Fused kinase, which has previously been taken to reflect the degree of inhibition by Su(fu), can be measured in *fu^{mH63}* wing discs. It can also be assayed in clonal patches of anterior cells (where Fu kinase is not activated), induced by mitotic recombination at earlier larval stages, to be homozygous for recessive mutations, such as *pka-C1* and *cos2*, to block proteolytic processing. Both *cos2* and *pka* mutant anterior clones induce high levels of Ci-155 (Fig 1F"), intermediate levels of *ptc-lacZ* (Fig 1F'), and little or no En [16]. In a *Su(fu)* mutant background, *ptc-lacZ* is increased to maximal levels (Fig 1G') and En is strongly induced in these clones (S1B Fig), while Ci-155 levels are substantially reduced (Fig 1G") [18].

## Activation and Su(fu) inhibition of Ci variants

Su(fu) binds directly to a conserved N-terminal region of Ci/Gli centered on an SYGHI/L motif (residues 255–259) [68,69]. Alteration of the SYGHI motif in Ci to SYAAD blocks Su(fu) binding to Ci fragment 230–272 in vitro [69]. Su(fu) also binds, albeit less avidly according to equivalent co-immune precipitation assays using restricted portions of Ci, to the extreme C-terminus of Ci-155 (1,370–1,397) and to at least one other, ill-defined site within residues 620–1,020 [70]. Su(fu) associ-ation with full-length Ci in co-immune precipitation assays is stronger when all binding sites are present, implying coopera-tive binding [70]. We constructed designer *ci* alleles using CRISPR to ask which Ci regions are important for physiological inhibition by Su(fu). The location of Ci domains and Ci variants examined in this study are summarized in S1A Fig and S1

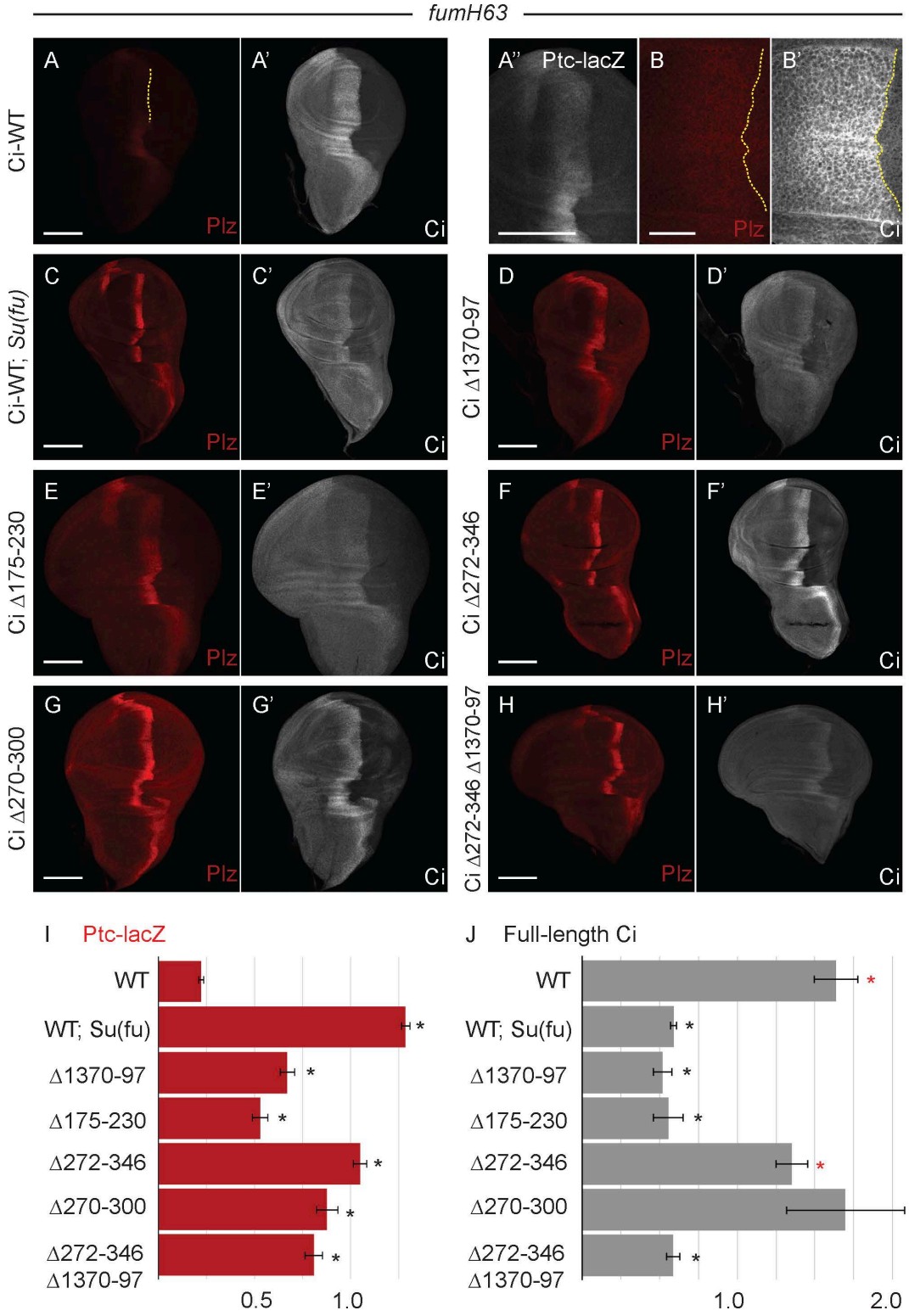

**Fig 2. Deletion of residues 270–300 increases Ci activity without loss of Su(fu)-dependent protection from degradation. (A–H)** Third instar *fu^mH63^* wing discs with one copy of the indicated Ci variant showing *ptc-lacZ* expression (Ptc-lacZ; red) and **(A'–H')** full-length Ci (Ci-155; gray-scale). **(B–B')** taken at higher magnification (63× vs. 20× objective for others). **(A, A", B)** Ptc-lacZ is greatly reduced by the loss of Fu kinase activity but expands

further anterior in the absence of Fu kinase activity (lower Ptc protein levels allow Hh to travel further); **(A")** shows the same Ptc-lacZ image shown in **(A)** but zoomed in and in gray-scale with contrast adjusted to show the width of weak expression. **(A', B')** High Ci-155 levels extend further anterior and right up to the AP border (yellow dotted line) in the absence of Fu kinase. **(C–C')** Ptc-lacZ is increased, with both Ptc-lacZ and Ci-155 stripes narrowed in the additional absence of Su(fu). **(D–H)** Ptc-lacZ was also increased, to a lesser degree, by the tested Ci alterations, together with narrowing of the widths of the elevated Ptc-lacZ and Ci-155 stripes. **(C'–H')** Ci-155 levels were reduced substantially by loss of Su(fu) and for the Ci variants shown, other than for Ci Δ272-346 and Ci Δ270–300. Scale bars are **(B, B')** 20 μm and (all others) 100 μm. **(I, J)** Bar graphs showing the average ratio of **(I)** Ptc-lacZ intensity at the AP border of $fu^{mH63}$ discs relative to the AP border of $fu^{mH63}$; Su(fu)$^{LP/LP}$ discs with one copy of Ci-WT, or **(J)** Ci-155 intensity relative to $fu^{mH63}$; Su(-fu)$^{LP/LP}$ wing discs expressing Ci-WT, and then adjusted to the wild-type wing disc control value, together with SEM values ($n$-values 55, 43, 13, 10, 18, 8, and 9, respectively, for each graph). Differences with $p < 0.005$ (Student $t$ test with Welch correction) are indicated for comparing a Ci variant to **(I, J)** Ci-WT (black asterisk) or **(J)** Ci Δ1,370–1,397 (red asterisk). Wing discs for variants with deletions neighboring the SYGHI Su(fu) binding site variably showed enlargement of the anterior compartment. This likely reflects impaired function of the processed repressor derivative, Ci-75, leading to localized de-repression of $dpp$ [16,57]. Please see Materials and methods for details of measurements and expression of all experimental values relative to AP border values of control wild-type wing discs, and S2 Data for raw data.

Table. The methods used to measure Ptc-lacZ and Ci-155 in anterior clones and at the AP border are fully described in Materials and methods; all values were normalized to control wing discs from the same experiment. Results are presented graphically, supported by raw and processed data, with statistics, in S1–S8 Data.

Both Ci with an altered SYGHI motif (Ci-SYAAD) and Ci lacking residues 1,370–1,397 induced $ptc$-$lacZ$ to much higher levels than Ci-WT in $cos2$ mutant clones, close to levels induced by Ci-WT in the complete absence of Su(fu) (Fig 1H, 1I, and 1N). Thus, assuming these relatively small perturbations do not have additional unexpected consequences, we conclude that Su(fu) must bind to both sites (SYGHI and 1,370–97) on a Ci-155 molecule for effective silencing.

From prior biochemical tests, we expected that residues 230–272 may constitute the whole Su(fu) binding domain around the SYGHI motif [69]. Surprisingly however, Ci lacking residues 175–230 or 272–346 also induced much higher levels of $ptc$-$lacZ$ than Ci-WT (Fig 1J, 1K, and 1N). Subsequent tests showed the same phenotype for deletion of residues 270–300, with no increase from deleting residues 301–320 or 321–346 (Fig 1L and 1N). Deletion of residues 175–230, 270–300, 272–346, or 1,370–1,397 also substantially increased $ptc$-$lacZ$ expression at the AP border of $fu^{mH63}$ discs relative to Ci-WT, consistent with increased Ci activity (Fig 2A–2G and 2I).

These results raised the question of whether residues neighboring 230–272 contribute directly to a Su(fu) binding interface. Ci-155 levels provided further information. The SYAAD alteration or deletion of residues 1,370–1,397 greatly reduced Ci-155 levels in $cos2$ mutant clones (Fig 1H, 1I, and 1O), consistent with an expectation that these variants have reduced stability because of reduced binding to Su(fu). In contrast, Ci-155 levels were much higher for Ci Δ272–346 and CiΔ270–300, and marginally higher for Ci Δ175–230 (Fig 1J, 1K, 1L, and 1O). A potential confounding factor is that En induction in these clones could also reduce Ci-155 levels by reducing $ci$ transcription. However, En, like $ptc$-$lacZ$, was induced more strongly by Ci Δ175–230, Ci Δ272–346, and Ci Δ270–300 than by Ci Δ1370–97 (S1 Fig). Thus, the higher Ci levels in $cos2$ mutant clones for the Ci variants lacking regions 175–230, and especially 270–300, likely report greater Ci stability than for Ci deficient for a direct Su(fu) contact.

None of the Ci variants tested in $fu^{mH63}$ mutant wing discs induced En at the AP border. Hence, Ci-155 levels should report Ci stability without any activity-dependent feedback on $ci$ transcription. Ci-155 levels at the AP border were very high, as for Ci-WT, for Ci lacking residues 270−300 or 272−346, while loss of residues 1,370−1,397 or 175−230 or Su(fu) absence each produced much lower Ci-155 levels (Fig 2A'–G' and 2J). Thus, Ci lacking residues 270−300 showed a strong increase in Ci activity in the presence of Su(fu) and the absence of Fu kinase, without the associated reduction in Ci-155 levels seen for Ci variants with alterations directly affecting Ci–Su(fu) interfaces.

One possibility was that the 270–300 region may directly facilitate Su(fu) binding but also contribute to a degron that de-stabilizes Su(fu)-free Ci-155. However, Ci lacking both residues 272–346 and 1,370–97 had low Ci-155 levels in $cos2$ mutant clones (Fig 1L" and 1O) and $fu^{mH63}$ mutant wing discs (Fig 2H' and 2J), similar to loss of 1,370–1,397 alone. Thus, residues 270–300 do not appear to be required for degradation of Su(fu)-free Ci-155. We therefore conclude that loss of residues 270–300 activates Ci without significantly disrupting Su(fu) binding. This shows that Ci can adopt a more active

conformation without altering the direct actions of Su(fu) and suggests that normal Ci activation by Hh may substantially reflect conversion of Ci to a more active conformation rather than directly countering inhibition by Su(fu).

To explore the activation state of Ci variants, further we tested their activities in anterior clones that expressed *UAS-GAP-Fu*, which produces activated Fu kinase [43,47]. These clones were generated in a way that simultaneously blocked *smo* activity (*smo UAS-GAP-Fu* clones; designated *GAP-Fu* clones for simplicity), so we can be certain we are measuring downstream consequences of Fu kinase activity in the absence of any potential Smo contribution. This assay contrasts with Fu responses measured at the AP border, where there are significant additional changes induced by activated Smo. GAP-Fu induced modest *ptc-lacZ* expression (30% of maximal AP border levels) for wild-type Ci and a small increase in Ci-155 relative to surrounding anterior cells, significantly less (42%) than maximal AP border levels (Fig 3A, 3G, and 3H). The increase in Ci-155 levels caused by expression of GAP-Fu is thought to reflect reduced proteolytic processing in response to Fu kinase activity, but the mechanism is not known [43]; this likely contributes, together with activation of Ci-155, to *ptc-lacZ* induction by GAP-Fu.

All of the Ci variants described above showed induction of *ptc-lacZ* to at least the levels of Ci-WT (30% of wild-type AP border levels) and modest elevation of Ci-155 within GAP-Fu clones, compared with surrounding anterior cells (Fig 3). Interestingly, *ptc-lacZ* induction was strikingly high for Ci lacking residues 270–300 (72% of control AP border level) and 272–346 (61%). Another activated Fu variant ("Fu-EE") activates wild-type Ci more strongly than GAP-Fu in this assay, so we deduce that not all Ci molecules are being phosphorylated at all relevant sites by GAP-Fu [43]. Hence, it is expected that a partially activated Ci variant will attain higher activity than Ci-WT. The observation that Ci Δ270–300 and Ci Δ272–346 exceed the activities of Ci-SYAAD (52%) and Ci Δ1,370–1,397 (30%) confirms the conclusion that their high activity cannot result from a minor reduction in Su(fu) binding (Fig 3G). It also suggests the possibilities that loss of residues 270–300 may allow more efficient phosphorylation by Fu (because of a more open, active conformation) or that it synergizes better with GAP-Fu because it mimics the normal activation state better than synthetic disruption of Ci–Su(fu) interfaces.

## Ci covalently linked to Su(fu) distinguishes among some models of Ci inhibition and activation

It has been suggested that Sufu anchors Ci/Gli in the cytoplasm [66,71–73], recruits co-repressors in the nucleus [74,75], or prevents binding of CBP co-activator and nuclear transport factors to Ci [70,76], and that Hh promotes Sufu dissociation from Ci/Gli [48,63,77]. We aimed to distinguish some of these possibilities by linking Su(fu) to Ci covalently. A *ci* allele ("Ci-WT-Sufu") was constructed to encode a fusion protein with full-length Su(fu) connected to the C-terminus of full-length Ci through a (GGGGS)$_3$ poly-Gly linker.

Ci-WT-Sufu induced only very low levels of *ptc-lacZ* in *cos2* clones and there was no significant increase in the absence of Su(fu) (Fig 4D, 4F, and 4L). We conclude that the Su(fu) moiety in Ci-WT-Sufu, rather than free Su(fu), is limiting Ci-155 activity. We then altered the SYGHI Su(fu) binding site to SYAAD within Ci-Sufu. Induction of *ptc-lacZ* by Ci-SYAAD-Sufu in *cos2* mutant clones was much greater than for Ci-WT-Sufu; it was not increased by removal of Su(fu), suggesting again that only tethered Su(fu) is participating in Ci inhibition (Fig 4E and 4G). Similar results were seen in *pka* mutant clones (S2A–S2D Fig). These results suggest that strong inhibition of Ci-155 by Su(fu) cannot be enforced simply by connecting the two molecules but requires Su(fu) to associate directly with the SYGHI-containing binding site and potentially other normal interaction sites.

The results also show that one (tethered) Su(fu) molecule can suffice to silence one Ci-155 molecule. Since the tethered Sufu moiety is always connected to Ci and unlikely to be less available for interaction with a cytoplasmic anchor or a nuclear co-repressor when SYGHI is substituted by SYAAD, it seems unlikely that inhibition by Su(fu) primarily involves binding to a cytoplasmic anchor or a co-repressor. The properties of these Ci-Sufu molecules are, however, compatible with Su(fu) shielding Ci from activity-promoting interactions with nuclear transport proteins or co-activators and with that shielding being promoted by non-covalent Ci–Su(fu) interactions, including at the SYGHI site. The Ci-Sufu molecules are

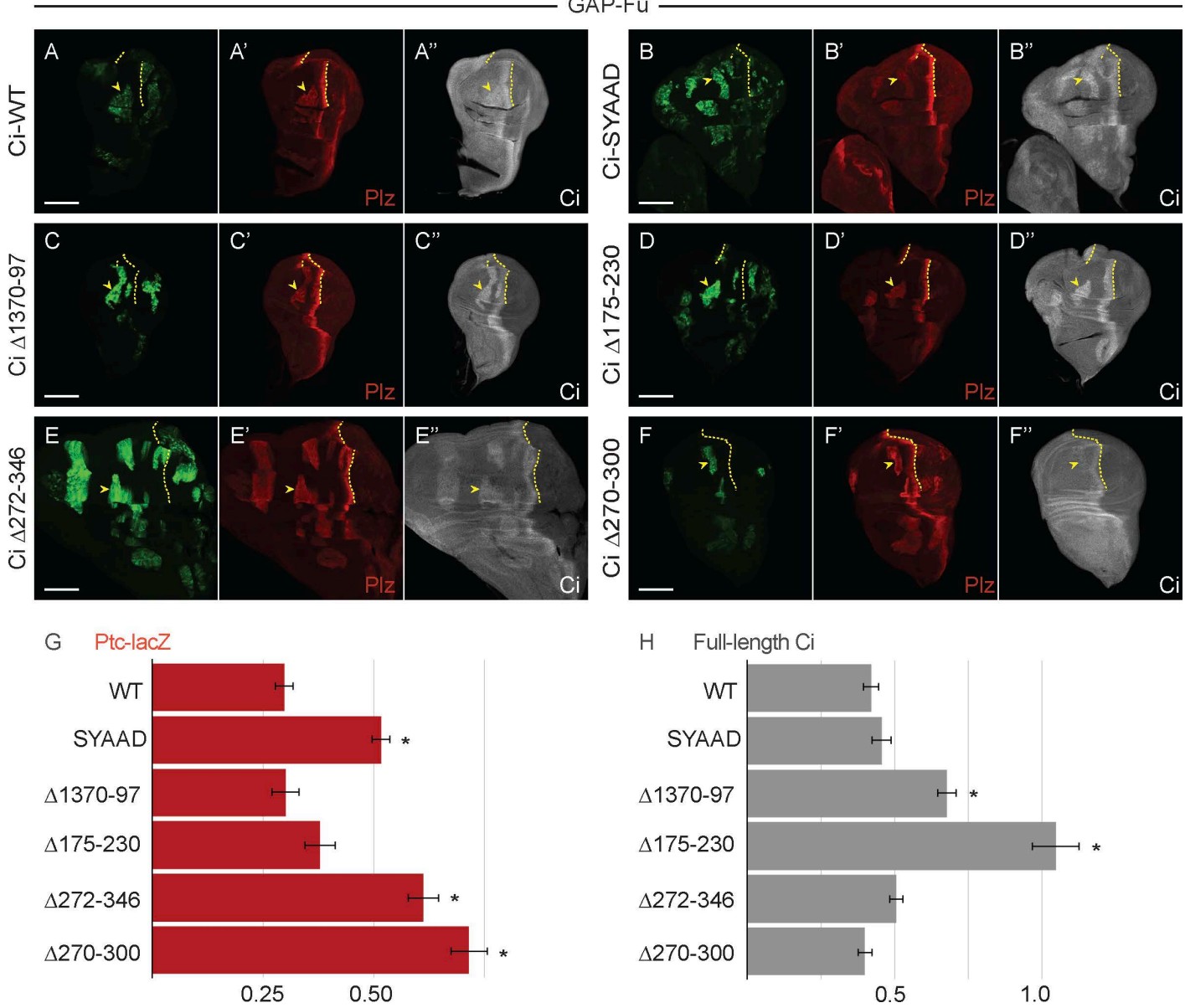

**Fig 3. Deletion of residues 270–300 confers potent activation by Fused kinase. (A–F)** Third instar wing discs (20× objective) with one copy of the indicated *ci* CRISPR allele, GFP marking homozygous *smo GAP-Fu* mutant clones (green; yellow arrowheads), and yellow dotted lines marking the AP border. **(A'–F')** Ptc-lacZ (red) and **(A"–F")** Ci-155 (gray-scale) in the same discs. Fu kinase activity in the clones increased Ptc-lacZ expression and Ci-155 levels in all cases, but Ptc-lacZ was induced most strongly by Ci Δ272–346 and Ci Δ270–300. Scale bars are 100 μm for all images. **(G, H)** Bar graphs showing the average ratio of **(G)** Ptc-lacZ intensity or **(H)** Ci-155 intensity in *smo* GAP-Fu clones relative to the AP border of wild-type control discs, together with SEMs (*n*-values 40, 9, 8 15, 19, and 20, respectively, for each graph). Differences with $p < 0.005$ (Student *t* test with Welch correction) are indicated for comparing a Ci variant to **(G, H)** Ci-WT (black asterisk). Please see Materials and methods for details of measurements and expression of all experimental values relative to AP border values of control wild-type wing discs, and S3 Data for raw data.

presumably more refractory than wild-type Ci to activation because an inactive Ci-Su(fu) complex is more stable in the presence of an additional, covalent, Ci-Su(fu) linkage.

We also tested the activation of Ci-Sufu proteins by activated GAP-Fu. Neither Ci-WT-Sufu nor Ci-SYAAD-Sufu induced detectable *ptc-lacZ* in response to GAP-Fu; Ci-155 levels increased slightly, similar to Ci-WT (Figs 4H, 4I, 4L, and S2E).

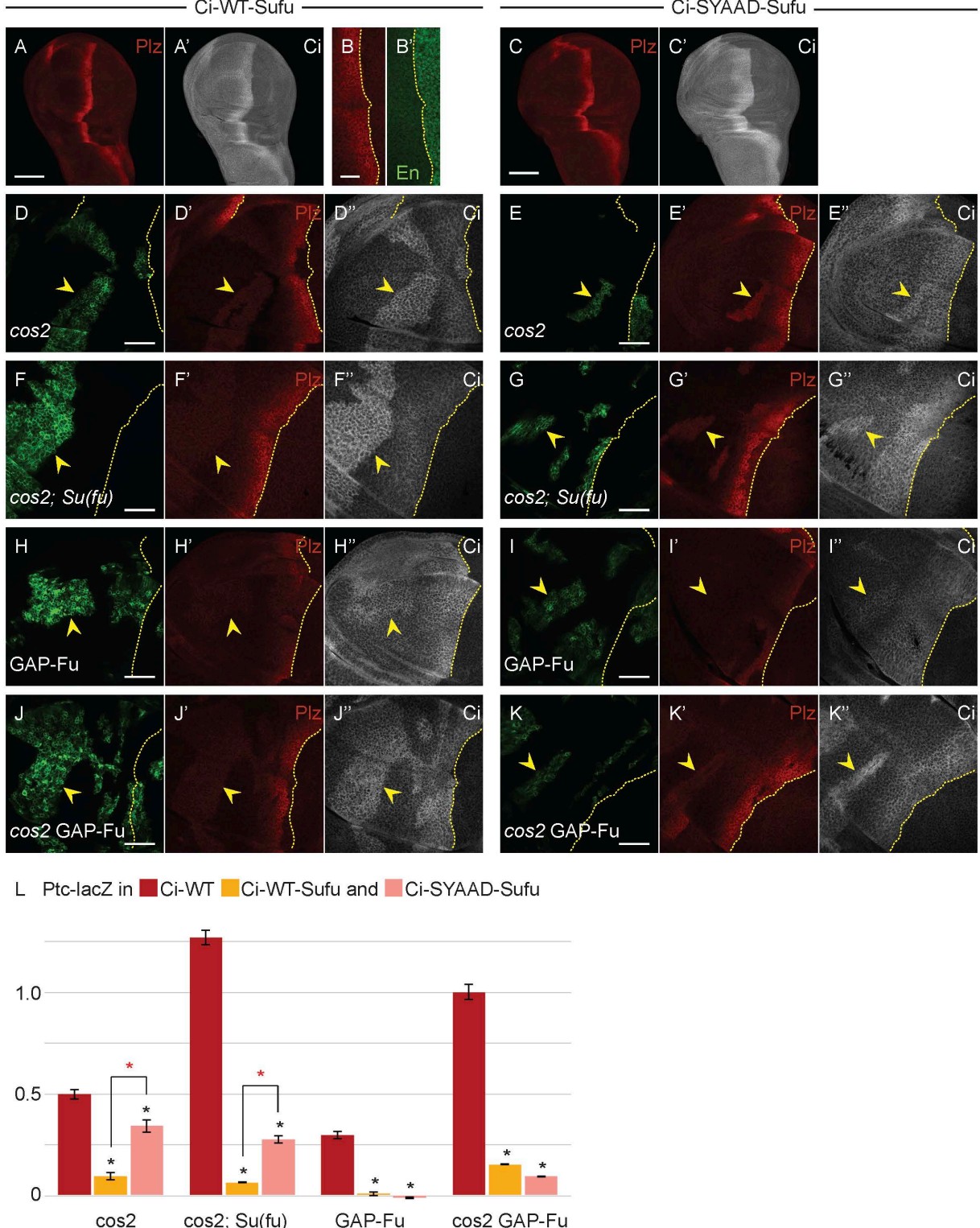

**Fig 4. Su(fu) inhibition is increased by covalent linkage to Ci but still requires non-covalent binding to the SYGHI region of Ci. (A, B)** Third instar wing disc with one copy of Ci-WT-Sufu (20× objective), showing **(A)** Ptc-lacZ (red) and **(A')** Ci-155 expression (gray-scale) or **(B)** Ptc-lacZ (red) and **(B')** En expression (green) at the AP border (yellow dotted line; 63× objective), showing no induction of anterior En. **(C)** Third instar wing disc with

one copy of Ci-SYAAD-Sufu (20× objective), showing Ptc-lacZ (red) and **(C')** Ci-155 expression (gray-scale). **(D–K)** Third instar wing discs (63× objective) with one copy of **(D, F, H, J)** Ci-WT-Sufu or **(E, G, I, K)** Ci-SYAAD-Sufu, GFP marking the indicated clone types (green; yellow arrowheads) and yellow dotted lines marking the AP border. **(D'–K')** Ptc-lacZ expression (red) and **(D"–K")** Ci-155 expression (gray-scale) in the same discs. **(F, G)** Su(fu) activity was absent in the whole disc. Ptc-lacZ induction in all clones was much lower for both Ci variants than for Ci-WT but was higher for Ci-SYAAD-Sufu. Scale bars are 100 µm for **(A, C)**, 20 µm for **(B)**, and 40 µm for **(D–K")**. **(L)** Bar graph showing the average ratio of **(L)** Ptc-lacZ intensity in clones relative to the AP border of wild-type control discs, together with SEMs (*n*-values 20, 5, 14, 10, 87, 19, 40, 54, 46, 47, 199, and 90). Differences with $p < 0.005$ (Student *t* test with Welch correction) are indicated for comparing a Ci variant to Ci-WT (black asterisk) or comparisons between bracketed pairs (red asterisk). Please see Materials and methods for details of measurements and expression of all experimental values relative to AP border values of control wild-type wing discs, and S4 Data for raw data. A bar graph of Ci-155 levels in the clones shown here for Ptc-lacZ, together with examination of *pka* mutant clones for Ci-WT-Sufu and Ci-SYAAD-Sufu, are shown in S2 Fig.

To potentially make this assay more sensitive, we tested clones that lacked *cos2* activity, as well as expressing GAP-Fu in the absence of *smo* activity. This produced maximal *ptc-lacZ* expression (100% of AP border level) and strong En induction for Ci-WT (Figs 4L, 5O, and S3I). In contrast, only a low level of *ptc-lacZ* expression was observed for Ci-WT-Sufu, roughly matching the levels in *cos2* clones, and *ptc-lacZ* expression was slightly lower than in *cos2* clones for Ci-SYAAD-Sufu (Fig 4J–L). Thus, Ci-Sufu molecules are extremely refractory to activation by Fu kinase alone.

Interestingly, in wild-type discs, Ci-Sufu proteins induced *ptc-lacZ* at the AP border to much higher levels (59% of wild-type AP border levels but without induction of anterior En; Fig 4A and 4B) than seen in *cos2* or *pka* mutant clones, where Ci processing is fully blocked (Ci-WT-Sufu induced Ptc-lacZ to 6% and Ci-SYAAD-Sufu to 16% of control AP border levels in *pka* mutant clones, S2A–D Fig). Ci-SYAAD-Sufu was also activated more strongly at the AP border than in *smo GAP-Fu* or *smo cos2 GAP-Fu* clones (Fig 4C). These results suggest that specific Hh-induced changes, present at the AP border but not in *smo GAP-Fu* or *smo cos2 GAP-Fu* anterior clones, collaborate with Fu kinase to activate Ci.

## Fu activation of Ci involves S218 and S1230 on Ci, but also additional sites

Past studies indicated that two specific Fu phosphorylation sites on Ci are critical for Ci activation, whereas identified Fu sites on Cos2 and Su(fu) are not [27,38,43,48,52]. The simplest expectation is therefore that substitution of the key Fu target sites on Ci with Ala or Val residues would abrogate all responses to Fu kinase to produce a phenotype identical to that of loss of Fu kinase activity. However, wing discs for Ci variants with Ala substitutions centered on the key Fu sites at S218 ("A1": S218A, S220A), S1230 ("A2": S1229A, S1230A, T1232V, S1233A) or both ("A1A2") all showed much stronger *ptc-lacZ* expression at the AP border than elicited by wild-type Ci in *fu^{mH63}* wing discs (Fig 5A, 5C, and 5E vs. Fig 2A and 2B; quantitation in S5M Fig), suggesting that these are not the only Fu sites that mediate Ci activation. Ci-A1 induced *ptc-lacZ* to wild-type levels, as well as inducing strong En, while *ptc-lacZ* expression was reduced for both Ci-A2, which retained weak En induction, and Ci-A1A2, which showed no En induction (Fig 5B', 5D', and 5F'). The induction of *ptc-lacZ* by Ci-A1A2 in wild-type discs (77% of normal AP border) was greatly reduced in *fu^{mH63}* mutant wing discs (22% of normal AP border) (Figs 5E, 5G, and S5N), providing direct confirmation that Fu kinase still activates Ci strongly in the absence of both S218 and S1230.

In contrast, the induction of *ptc-lacZ* by GAP-Fu in anterior cells was greatly reduced (and barely perceptible) for Ci-A1A2 compared with Ci-WT (Fig 5K vs. Fig 3A; quantitation in Fig 6J). The reduction was almost as strong for Ci-A2, but there was also a clear reduction for Ci-A1 (Figs 6J, S3A, and S3B). In all cases, the increase in Ci-155 levels within GAP-Fu clones, relative to neighboring cells, appeared similar to Ci-WT (Figs 5K, S3A, and S3B), suggesting that the mechanism responsible for reducing full-length Ci processing or degradation was unaltered by these S/A substitutions.

To explore why Ci-A1A2 is more readily activated by Fu at the AP border than in GAP-Fu clones, and to increase the sensitivity of the latter assay, we also tested the response to GAP-Fu in clones lacking *cos2*. In *cos2* clones, the induction of *ptc-lacZ* was slightly higher for Ci-A1A2 (59% of wild-type AP border levels) than for Ci-WT (50%) (Figs 5M, 5N, and 6K). The addition of GAP-Fu to *cos2* mutant clones increased *ptc-lacZ* for Ci-WT (to 100%, plus En induction) but decreased *ptc-lacZ* expression for Ci-A1A2 (34%), without altering Ci-155 levels (Figs 5O, 5P, S3I, and 6K). The failure to increase Ci

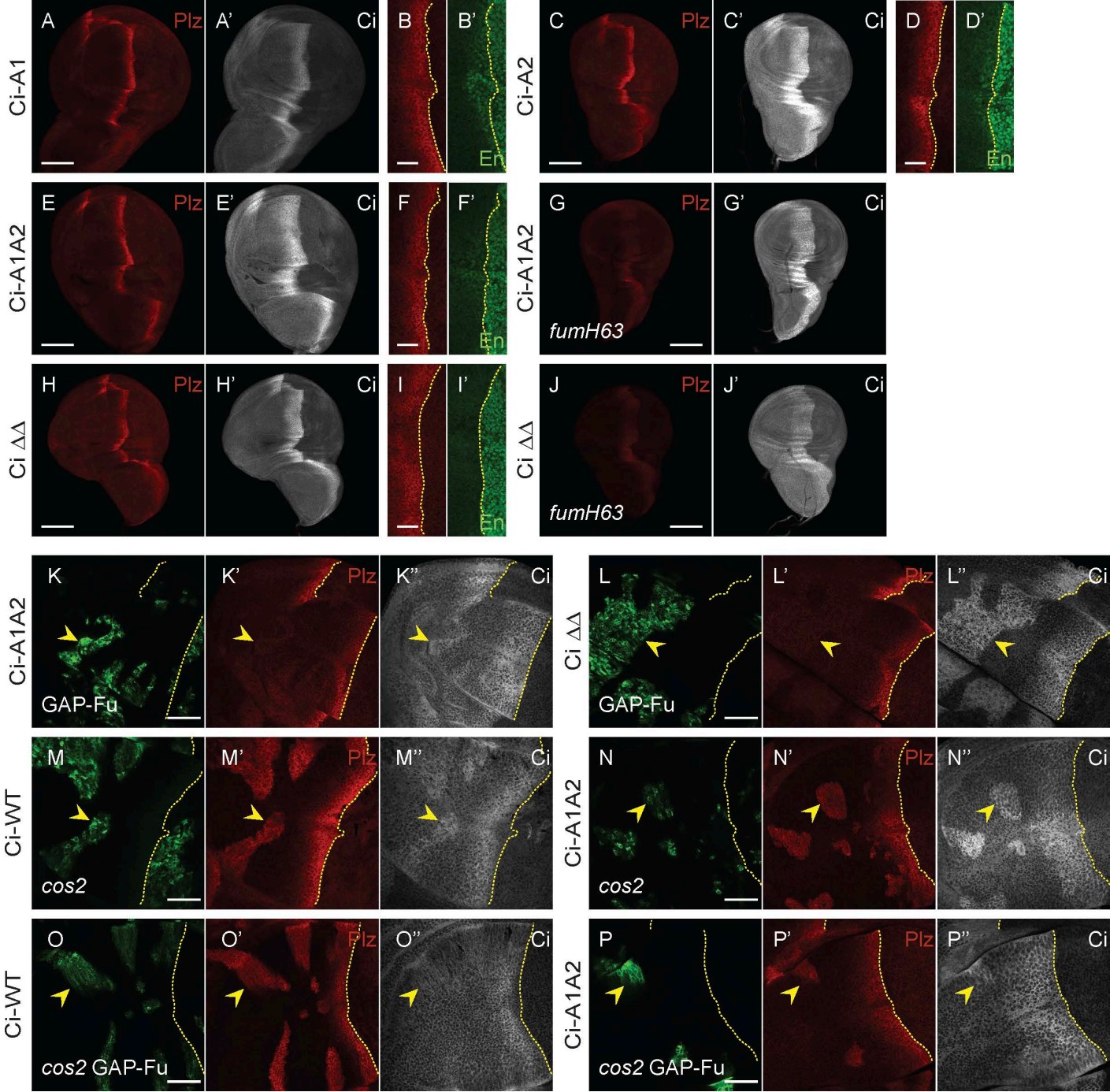

**Fig 5. Loss of S218 and S1230 Fu sites reduces Ci activation by Fu more severely in anterior cells than at the AP border. (A–J)** Third instar wing discs with one copy of each Ci variant showing **(A, C, E, H)** Ptc-lacZ (red) and **(A', C', E', H')** Ci-155 expression (gray-scale) (20× objective) or **(B, D, F, I)** Ptc-lacZ (red) and **(B', D', F', I')** En expression (green) (63× objective), with the AP border marked by dotted yellow lines. Induction of Ptc-lacZ was unchanged for **(A)** Ci-A1 and reduced progressively more by **(C)** Ci-A2, **(E)** Ci-A1A2, and **(H)** Ci ΔΔ (which lacks residues 175–230 and 1,201–1,271). En induction was normal for **(B')** Ci-A1, weak for **(D')** Ci-A2 and absent for **(F')** Ci-A1A2 and **(I')** Ci ΔΔ. **(G, J)** $fu^{mH63}$ discs with one copy of the indicated Ci variant showing Ptc-lacZ (red) and **(G', J')** Ci-155 expression (gray-scale). **(K–P)** Third instar wing discs (63× objective) with one copy of the named Ci variant, GFP marking the indicated clone types (green; yellow arrowheads), and yellow dotted lines marking the AP border. **(K'–P')** Ptc-lacZ expression (red) and **(K''–P'')** Ci-155 expression (gray-scale) in the same discs. Ptc-lacZ (red) was not induced by activated Fu kinase for **(K')** Ci-A1A2 or **(L')**

Ci ΔΔ, even in the absence of *cos2* activity (compare **O'** and **P'** to **M'** and **N'**; see bar graphs in Fig 6). **(K"–P")** Ci-155 was elevated similarly for all Ci genotypes by GAP-Fu and by loss of Cos2. Scale bars are 100 µM for **(A, C, E, G, H, J)**, 20 µm for **(B, D, F, I)**, and 40 µm for **(K–P)**. Quantitative results are included in the graphs of Fig 6 for **(K–P)** and S5 Fig for **(A–I)**. Please see Materials and methods for details of measurements and expression of all experimental values relative to AP border values of control wild-type wing discs, and S5 Data and S8 Data for raw data. See S3 Fig for *GAP-Fu* clones with Ci-A1 and Ci-A2; Ci Δ1201–1,271 (wild-type discs, *fu^{mH63}* discs, *cos2,* and *GAP-Fu* clones); *cos2 GAP-Fu* clones (Ci-WT, Ci ΔΔ, Ci Δ175–230, Ci Δ270–300, Ci Δ1370–1,397).

activity in the absence of proteolytic processing contrasts starkly with the impact of Fu kinase on Ci-A1A2 at the AP border. We deduce that Cos2 may facilitate activation of Ci-A1A2 by Fu and may do so most effectively at the AP border.

We also tested a Ci variant lacking residues 1,201–1,271, which includes the S1230 Fu site. It induced *ptc-lacZ* to near-normal levels in GAP-Fu clones (Figs 6J and S3G) and supported normal *ptc-lacZ* and normal En induction at the AP border (S3C, S3D, and S5N Figs). Induction of *ptc-lacZ* was also similar to Ci-WT in *cos2* mutant clones (S3F Fig) and in *fu^{mH63}* mutant wing discs (S3E and S5N Figs). Thus, the deletion mutant phenotype does not suggest reduced Ci activation by Fu kinase, in contrast to the properties of Ci-A2. However, when this deletion was combined with loss of residues 175–230 to create "Ci ΔΔ", there was, as for Ci-A1A2, barely any induction of *ptc-lacZ* in GAP-Fu clones (8%) (Figs 5L and 6J), consistent with the deduction that a strong response to GAP-Fu requires phosphorylation of S218 or S1230. Ci ΔΔ also produced a significant AP border reduction of *ptc-lacZ* and no En induction (Figs 5H, 5I, and S5N)—a slightly greater activity deficit than for Ci-A1A2. Ci ΔΔactivity at the AP border was greatly reduced by loss of Fu kinase activity, as for Ci-A1A2 (Figs 5H, 5J, and S5N).

The induction of Ptc-lacZ in *cos2* clones was reduced by GAP-Fu for both Ci Δ175–230, (from 81% to 52% of control AP border levels) and even more substantially for Ci ΔΔ (from 87% to 29% of control AP border levels) (Figs 6K, S3H, S3J, and S3L). Thus, a negative effect of GAP-Fu within *cos2* clones was observed in all Ci variants tested where S218 was absent (Ci-A1A2, Ci Δ175–230, and Ci ΔΔ) (Figs 5, S3, and 6K).

## Phospho-mimetic alterations around S218 and S1230 do not suffice to activate Ci

If phosphorylation of Ci at residues S218 and S1230, followed by phosphorylation at surrounding potential primed CK1 sites, were sufficient for Ci activation or escape from Su(fu) inhibition, then acidic residue substituents at these sites might mimic those properties. Following the design of Han and colleagues [48], we made *ci* alleles encoding the changes S218D, S220D, S1229D, S1230D, T1232E, and S1233D ("Ci-D1D2"). Induction of *ptc-lacZ* in *cos2* mutant clones was only slightly greater than for Ci-WT (57% of control AP border vs. 50%) and no greater than for Ci-A1A2 (59%) (Fig 6A and 6K). Induction of *ptc-lacZ* at the AP border of *fu^{mH63}* wing discs was higher than for Ci-WT (35% vs. 17% of control AP border levels) (Figs 6B and S5N) and Ci-A1A2 (22%) but much lower than observed for removal of Su(fu) from otherwise normal *fu^{mH63}* wing discs (129%; Fig 2). Thus, Ci-D1D2 was still substantially inactive in the absence of Fu kinase activity.

In wild-type discs expressing Ci-D1D2, the AP border pattern and levels of expression of *ptc-lacZ* and En were normal, matching controls (Fig 6C). This suggests that the acidic substituents are reasonably good functional mimics of phosphorylation. An exact assessment is not possible because the stoichiometry of normal phosphorylation is unknown, albeit likely to be less than complete. The difference between wild-type and *fu^{mH63}* discs at the AP border for Ci-D1D2 already indicates that this Ci variant can still be activated by Fu kinase activity. We also tested this directly in GAP-Fu clones. Induction of *ptc-lacZ* was similar to Ci-WT (28% vs. 30% of control AP border levels), with a similar modest elevation of Ci-155 (Fig 6D and 6J). Furthermore, GAP-Fu promoted stronger *ptc-lacZ* induction in *cos2* mutant clones (94% vs. 57% of control AP border levels), similar to the phenotype of Ci-WT (Fig 6E and 6K). Thus, there are clearly very influential additional Fu sites other than S218 and S1230, which can activate Ci-D1D2 in response to Fu kinase activity. Those same sites are available also in Ci-A1A2, for which Fused kinase activity leads to reduced activity of Ci within *cos2* clones, very little activation of Ci in GAP-Fu clones and significant but incomplete Ci activation at the AP border. Hence, the activation of Ci promoted by these additional, unknown Fu sites is only robustly realized if there is also some phosphorylation (or acidic residue mimics) at the S218 or S1230 regions.

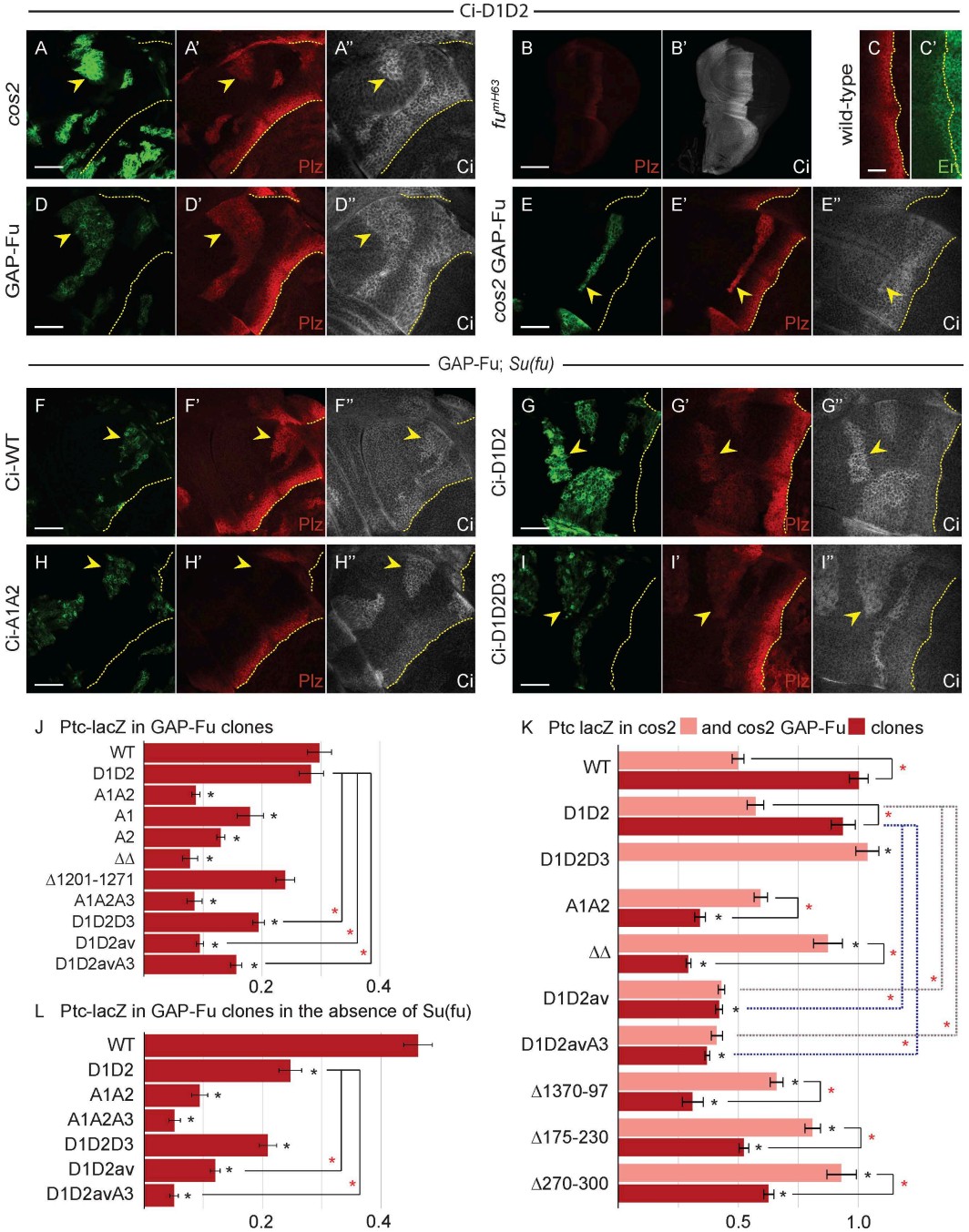

**Fig 6. S218 and S1230, as well as other Fu sites contribute to Ci activation even in the absence of Su(fu). (A, D–I)** Third instar wing discs (63× objective) with **(A, D, E)** one copy of Ci-D1D2 or **(F–I)** the named Ci variant, GFP marking the indicated clone types (green; yellow arrowheads), and yellow dotted lines marking the AP border. **(F–I)** shows *smo GAP-Fu* clones in *Su(fu)* mutant discs. **(A', D'–I')** Ptc-lacZ expression (red) and **(A'', D''–I'')** Ci-155 expression (gray-scale) in the same discs. **(B)** *fu^{mH63}* disc with one copy of Ci-D1D2, showing **(B)** Ptc-lacZ (red) and **(B')** Ci-155 expression (gray-scale) (20× objective). Wild-type disc with Ci-D1D2, with **(C)** Ptc-lacZ (red) and **(C')** En expression (green) at the AP border (yellow dotted line; 63× objective). Scale bars are 100 µm for **(B)**, 20 µm for **(C)**, and 40 µm for all other images. **(J, L)** Bar graphs showing the ratio of Ptc-lacZ intensity in *smo GAP-Fu* clones in **(J)** otherwise wild-type or **(L)** Su(fu) mutant discs relative to the AP border of control discs, together with SEMs ($n = 40, 31, 113, 10, 72, 28, 15, 25, 120, 55$, and $14$, respectively, in **(J)** and $79, 70, 45, 30, 75, 99$, and $35$, respectively, in **(L)**). **(K)** Bar graph showing the ratio of Ptc-lacZ intensity in *cos2* clones (pink) and in *smo cos2 GAP-Fu* clones (red) relative to the AP border of control discs, together with SEMs ($n = 20, 38, 45, 67, 38, 55, 30, 22, 53$, and $36$, respectively, for *cos2* and $47, 49, 7, 112, 75, 104, 15, 74$, and $38$, respectively, for *smo cos2 GAP-Fu*). **(J–L)** Differences with

*p* < 0.005 (Student *t* test with Welch correction) are indicated for comparing a Ci variant to Ci-WT (black asterisk) or comparisons between bracketed pairs (red asterisk). Gray dotted brackets compare *cos2* clones and blue dotted brackets compare *cos2 GAP-Fu* clones for Ci-D1D2. Please see Materials and methods for details of measurements and expression of all experimental values relative to AP border values of control wild-type wing discs, and S5 Data for raw data.

## How is Ci activity modulated by Fu phosphorylation?

The activity of Ci-155 can be limited or opposed by proteolytic processing or by binding to Cos2 or Su(fu). Thus, Ci-155 activation by Fu kinase could involve countering any or all of these inhibitory processes. The stimulation of Ci-D1D2 activity by GAP-Fu in *cos2* clones might therefore result from escaping Su(fu) inhibition since there is no proteolytic processing of Ci-155 and no Cos2 to restrain Ci-155 activity in these clones. Likewise, the failure of Ci-A1A2 activation by GAP-Fu under the same conditions may reflect an inability to escape Su(fu) inhibition (Fig 6K). Thus, both these known (S218 and S1230) and unknown Fu targets may contribute to escape from Su(fu) inhibition. Consistent with this possibility, Ci-A1A2 wing discs had near normal *ptc-lacZ* and some weak En induction (which was stronger with two copies of the Ci variant) at the AP border when Su(fu) was absent (quantitation in S5M Fig).

Although Fu kinase is commonly described as principally opposing Ci inhibition by Su(fu), it was previously deduced that Fu kinase must also activate Ci in another way, based on the absence of anterior En at the AP border of *fu^mH63^; Su(fu)* mutant discs [43]. A caveat to the implication of a direct Su(fu)-independent action of Fu on Ci is evidence that Fu kinase contributes to full Smo activation at the AP border [78], potentially leading indirectly to Ci activation.

To clarify this important issue and test the relevance of specific Fu targets for any actions independent of Su(fu), we examined GAP-Fu clones (still lacking *smo* activity) in a *Su(fu)* mutant background. Slightly elevated Ci-155 was seen in clones for Ci-WT, Ci-D1D2, and Ci-A1A2 (Fig 6F"–H"). Thus, the mechanism responsible for this elevation does not appear to involve Su(fu), consistent with the proposed mechanism of reduced proteolytic processing. There was no *ptc-lacZ* expression in anterior cells outside the clones, but it was clearly induced by GAP-Fu for Ci-WT (46% of control AP border level) and Ci-D1D2 (25%) (Fig 6F and 6G). There was virtually no *ptc-lacZ* induction for Ci-A1A2 (measured as 10%) (Fig 6H and 6L). The induction of *ptc-lacZ* could be due to both Ci-155 activation and increased Ci-155 levels. However, the increase in Ci-155 levels was no greater for Ci-WT or Ci-D1D2 than for Ci-A1A2 (Fig 6F"–H"). We therefore deduce that both full-length Ci-WT and Ci-D1D2 are being significantly activated by GAP-Fu, while Ci-A1A2 remains largely inactive. Hence, both the known (S218 and S1230) and unknown Fu targets can contribute to Ci activation independent of Su(fu), via a mechanism that does not involve elevation of full-length Ci levels.

One possibility is that Fu kinase opposes Ci inhibition by Cos2. Another possibility is that Fu promotes a change to a more active Ci conformation that is largely independent of other binding partners. In essence, this translates to the possibility of internal Ci–Ci interactions being modified directly by Fu phosphorylation of Ci. This explanation is highly favored because it is consistent with the observations that alterations to S218 and S1230 affected Fu stimulation of Ci activity in the absence of Su(fu), in the absence of Cos2, and in the presence of both. Thus, we propose that Fu kinase overcomes inhibition by Su(fu) principally by promoting an active Ci conformation rather than by directly opposing the inhibitory effect of Su(fu) binding. The proposed mechanism does not exclude additional direct effects of Su(fu) or Cos2 binding on Ci activity, for example through steric hindrance. Moreover, Ci binding partners, Su(fu) or Cos2, may also influence conformational transitions through differential affinity for different Ci conformers.

## Testing Cos2 and Ci phosphorylation requirements for Fu kinase to increase Ci-155 levels

Fu kinase activity is not required for Hh to inhibit Ci-155 proteolytic processing at the AP border, based on Ci-155 levels [18,45]. A sensitive test for repressor activity conducted in posterior cells provided further evidence that processing can be blocked completely in the absence of Fu kinase [27]. Hence, it was initially surprising to find that activated Fu (GAP-Fu

or Fu-EE) could increase Ci-155 levels in anterior *smo* mutant clones [43]. Suppression of the increase by excess PKA argued that the increase was due to reduced proteolytic processing, which is normally promoted by PKA. It was also found that the increase in Ci-155 was seen when wild-type Cos2 was over-expressed (using *C765-GAL4* plus *UAS-Cos2*) in *smo cos2* clones expressing Fu-EE, but not when Cos2 with a S572A alteration was expressed analogously [43]. We wished to test the consequent conclusion that Cos2 S572 phosphorylation mediates Ci-155 stabilization by Fu kinase under conditions of physiological protein expression.

We therefore performed a similar experiment in *smo cos2* mutant clones, but expressing GAP-Fu (rather than Fu-EE) and using a genomic transgene to express normal ("gCos-WT") or variant Cos2 ("gCos-AA") at physiological levels (in one allelic dose). We chose to use the variant Cos2 with both S572A and S931A alterations (gCos-AA), and we performed the test using a variety of Ci molecules. In all cases, Ci-155 levels were modestly increased in clones compared with neighboring cells for wing discs expressing wild-type Cos2 (gCos-WT) (Fig 7A, 7C, and 7E). This difference was less obvious in the presence of Cos2-S572A S931A than for Cos2-WT and was not discernible at all in some clones (Fig 7B, 7D, and 7F). Measurements showed slightly lower Ci-155 levels for the Cos2 variant (gCos-AA) compared with wild-type Cos2 (gCos-WT) within clones as absolute levels (Fig 7G) and after subtracting Ci levels in anterior cells outside clones (Fig 7H). Thus, we infer that Ci-155 levels are increased slightly by allowing Cos2 S572 and S931 phosphorylation. For Ci-WT, induction of *ptc-lacZ* by GAP-Fu was also slightly reduced for Cos2-S572A S931A (42% of control AP border levels) compared with wild-type Cos2 (59%), consistent with slightly higher Ci-155 levels endowing slightly higher Ci activity. Thus, these tests provide some support for a contribution of Cos2 phosphorylation to the increase in Ci-155 levels induced by Fu kinase. However, there were many clear examples of GAP-Fu clones with elevated Ci-155 (compared with surrounding cells) in the presence of Cos2-S572A S931A, including for Ci-A1A2 and Ci ΔΔ, which lack the Fu sites S218 and S1230 on Ci (Fig 7D and 7F). Thus, there must be additional Fu targets that contribute to increasing Ci-155 levels.

**Responses to Fu kinase persist when Ci, Su(fu), and Cos2 targets are altered in combination**

The known Fu sites in Su(fu) were previously tested in a *UAS-Su(fu)* transgene with five Ser to Ala alterations ("Su(fu)5A") [43]. These sites are not conserved in vertebrate Sufu. It was also found that mouse Sufu coding region ("mSufu"), expressed analogously, could inhibit Ci at the AP border of *fu^mH63^; Su(fu)* wing discs and permit normal Ci activation in *Su(fu)* mutant wing discs [43]. We tested whether the reduced activity of Ci-A1A2 at the AP border could be exacerbated by using *UAS-Su(fu)5A* or *UAS-mSufu*, compared with *UAS-Su(fu)-WT*, each expressed throughout the wing disc using *C765-GAL4* in a *Su(fu)* mutant background. Using one copy of Ci-A1A2, *ptc-lacZ* induction at the AP border was marginally reduced by Su(fu)5A (from 77% of control AP border levels to 66%), but a similar reduction was seen for Su(fu)-WT (to 66%) and presumably therefore results simply from overexpression of Su(fu) (S4A, S4D, S4F, and S4L Fig). Reduced activity was also seen for two copies of Ci-A1A2 (loss of weak En induction and *ptc-lacZ* falling from 90% to 82% (Su(fu)5A) or 69% (Su(fu)-WT)) (S4B, S4C, S4E, S4G, and S4L Fig). mSufu did not reduce AP border activity of either one copy (*ptc-lacZ* 91% of control AP border) or two copies of Ci-AA (87% *ptc-lacZ* and En induction), likely reflecting a lesser inhibitory potential of mouse Sufu under these sensitized conditions (S4I–L Fig).

We then additionally modified the second chromosome to be homozygous for a loss of function *cos2* allele together with a genomic Cos2-S572AS931A transgene ("gCos-AA") [27] and tested the activity of two copies of Ci-A1A2. There was no reduction of *ptc-lacZ* in the presence of Su(fu)5A (88% of control AP border level) or mSufu (111%) (S4L–O Fig). All of the genotypes described above also yielded adult flies with wings of normal appearance. Thus, there were no major disruptions to molecular measures of Hh signaling at the AP border or the morphological functions of Hh signaling when the known Fu phosphorylation sites on Su(fu) and Cos2 were absent together with the two sites in Ci (S218 and S1230) reported to be the major functional Fu targets on the basis of earlier, non-physiological tests.

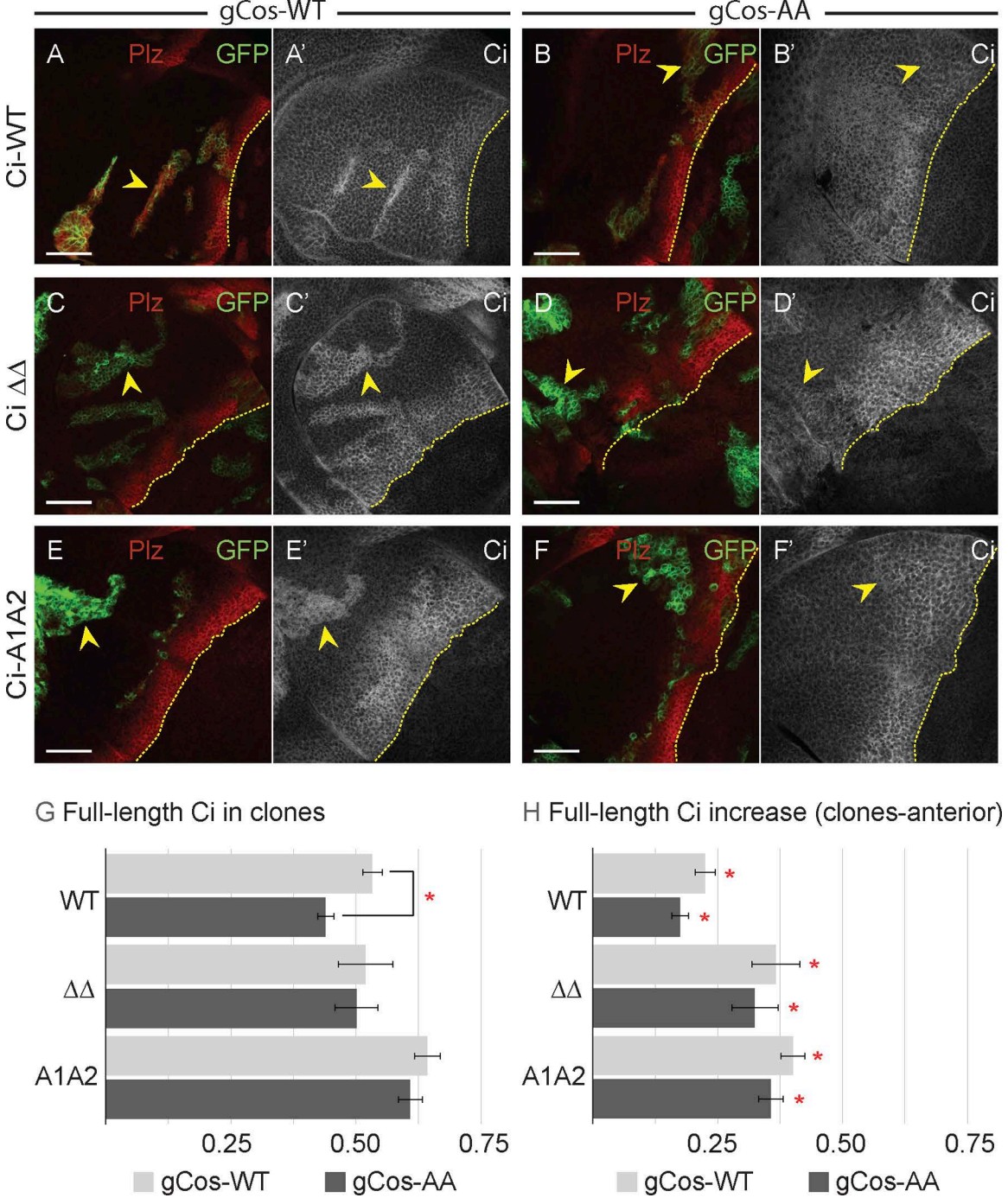

**Fig 7. Cos2 phosphorylation contributes to the elevation of full-length Ci protein by activated Fu kinase. (A–F)** Third instar wing discs with one copy of the named Ci variant (63× objective), showing GFP expression (green) to mark *smo cos2* GAP-Fu clones (yellow arrowheads) and *ptc-lacZ* (red) to mark the AP border in wing discs with one genomic copy of a transgene encoding **(A, C, E)** wild-type Cos2 (gCos-WT) or **(B, D, F)** Cos2 S572A S931A (gCos-AA). **(A'–F')** Ci-155 expression (gray-scale); AP border marked by yellow dotted lines on the basis of Ptc-lacZ expression (not shown). Ci-155 was clearly higher in all clones with gCos-WT but was more similar to surrounding anterior cells away from the AP border in the presence of gCos-AA, with the clearest distinction observed for **(F')** Ci-A1A2. Scale bars are 40 μm for all images. **(G)** Bar graph showing the ratio of Ci-155 intensity in *smo cos2* GAP-Fu clones relative to the AP border of wild-type control discs, together with SEMs (n = 116, 52, 21, and 96, respectively, for gCos-WT and 76, 38, 26, and 98, respectively, for gCos-AA) for the designated Ci variants together with gCos-WT (gray) or gCos-AA (black). **(H)** Bar graph

showing the difference in the ratio of Ci-155 intensity between *smo cos2* GAP-Fu clones and the surrounding anterior cells, relative to the AP border of wild-type control discs, together with SEMs (*n* values the same as **G**) for designated Ci variants together with gCos-WT (gray) or gCos-AA (black). **(G)** Differences with $p < 0.005$ (Student *t* test with Welch correction) are indicated for differences between values for gCos-WT vs. gCos-AA for a given Ci (red asterisk). **(H)** Differences with $p < 0.005$ (Student *t* test with Welch correction) are indicated for comparing the change in Ci-155 level induced by the clone genotype relative to zero (red asterisk). Please see Materials and methods for details of measurements and expression of all experimental values relative to AP border values of control wild-type wing discs, and S6 Data for raw data. See S5 Fig for the effect of loss of Fu sites in Su(fu) and Cos2 on the activity of Ci-A1A2.

## Two additional sets of Fu sites on Ci influence Ci activation

Two sites of Fu-induced phosphorylation (S286, T294) were initially dismissed as having little effect in an assay of Fu responses in cell culture [48] but lie within the region of a deletion (residues 270–300) that activated Ci in a manner that suggested a possible effect on Ci conformation rather than direct binding to Su(fu). We therefore altered these residues (S286A T294V) in the context of Ci-D1D2 to test the response to activated Fu. The induction of *ptc-lacZ* by GAP-Fu was reduced to minimal levels, similar to Ci-A1A2, suggesting that the S286/T294 sites are important targets for Ci activation by Fu (Figs 6J and 8M). This variant ("Ci-D1D2av") induced *ptc-lacZ* similarly to Ci-WT in *cos2* mutant clones (43% of control AP border level), but induction was not increased further by the addition of GAP-Fu (42%) (Figs 8O, 8P, and 6K), in contrast to Ci-WT or Ci-D1D2, revealing a clear deficit in the response to Fu away from the AP border, with or without Cos2. However, both *ptc-lacZ* and En were induced normally at the AP border (Fig 8Q and 8R), dependent on Fu kinase activity (Figs 8S and S5N), implying normal activation by Fu at this location. Ci Δ270–300 responded strongly to GAP-Fu (to give *ptc-lacZ* at 72% of control AP border levels), as described earlier, but GAP-Fu reduced activity in *cos2* mutant clones (93%–63%) (Figs 6K and S3M). Thus, although S286A and T294V have not been tested in isolation, it is clear that S286 and T294 phosphorylation can contribute to Fu kinase responses in some conditions but is not essential at the AP border. There was also virtually no induction of *ptc-lacZ* by Ci-D1D2av in GAP-Fu clones in the absence of Su(fu), showing that S286 and T294 phosphorylation can contribute to Ci activation independent of Su(fu) (Figs 8N and 6L).

A Fu phosphorylation site, initially missed by mass spectrometry studies and found later by directed tests, lies near the extreme C-terminus of Ci within a region (residues 1,370−97) sufficient for binding to Su(fu) [52]. Alteration of the S1382 Fu site and S1385 Fu-primed CK1 sites to Ala or acidic residues was reported to enhance the effects of corresponding changes of the S218 and S1230 regions in assays of Fu responses in tissue culture and consequent to ubiquitous additional expression of *ci* variant transgenes in wing discs [52]. We found that under physiological conditions, altering the C-terminal residues to Ala (Ci-A3: S1382A S1385A) did not affect induction of *ptc-lacZ* or En at the AP border (S5A, S5B, and S5M Fig). This additional change ("Ci-A1A2A3") did, however, reduce expression of *ptc-lacZ* relative to Ci-A1A2 (from 77% to 53% of control AP border levels) (Figs 8A, 8B, and S5M). In the absence of Su(fu), En was induced, along with normal *ptc-lacZ* (117% of control AP border levels) at the AP border for Ci-A1A2, whereas *ptc-lacZ* was not fully restored (85%) and there was no En induction for Ci-A1A2A3 (Figs 8C–F and S5M). There was very little response of Ci-A1A2A3 to GAP-Fu in the presence (9% of control AP border *ptc-lacZ* levels) or absence (5% of control AP border *ptc-lacZ*) of Su(fu), similar to Ci-A1A2, and GAP-Fu reduced *ptc-lacZ* induction in the absence of *cos2* (from 55% to 22% of AP border levels), also similar to Ci-A1A2 (Figs 6J, 6L, S5F, and S5G).

Adding the A3 alteration to a backbone of D1D2av (to make Ci-D1D2avA3) did not impair strong *ptc-lacZ* and En activation at the AP border, which was dependent on Fu kinase activity (S5J–L and S5N Fig). Ci-D1D2avA3 was less strongly activated (to 16% of control AP border levels) than Ci-WT (30%) and Ci-D1D2 (28%) by GAP-Fu, though slightly more than Ci-D1D2av (9%) (Figs 6J and S5H). It was not activated by GAP-Fu in *cos2* mutant clones (*ptc-lacZ* changed from 41% to 37% of control AP border levels) (Fig 6K) or in the absence of Su(fu) (Figs 6L and S5I). Thus, Ci-D1D2avA3 was largely refractory to activation by Fu kinase in anterior cells but was strongly activated at the AP border, suggesting that there must be important Fu targets other than S218, S1230, S286, T294, and S1382 of Ci.

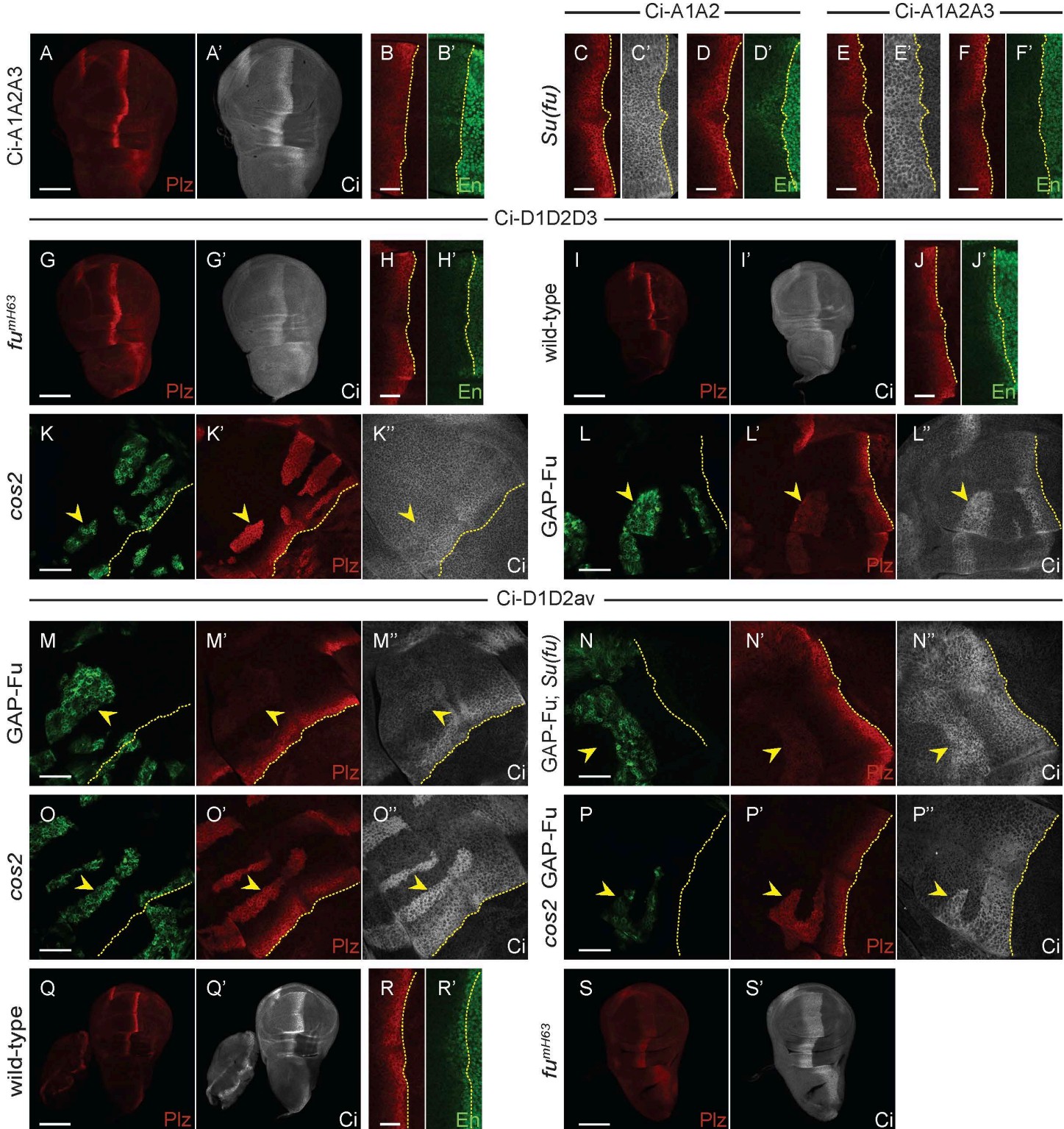

**Fig 8. Acidic substituents at three Fu sites substantially compensate for Fu kinase activity but additional sites contribute to the Fu response.**
**(A–F, I, J, R–T)** One copy of named Ci variants in **(A, B, E, F, I, J, Q, R)** otherwise wild-type, **(C, D)** *Su(fu)*^LP/LP^, or **(G, H, S)** *fu*^mH63^ third instar wing discs, showing **(A–J, Q–S)** Ptc-lacZ (red), **(A', C', E', G', I', Q', S')** Ci-155 (gray-scale), or **(B', D', F', H', J', R')** En expression (green, with the AP border

marked by dotted yellow lines. **(K–P)** Third instar wing discs (63× objective) with the named Ci variant, GFP marking the indicated clone types (green; yellow arrowheads), and yellow dotted lines marking the AP border. **(K'–P')** Ptc-lacZ expression (red) and **(K"–P")** Ci-155 expression (gray-scale) in the same discs. **(L, M, N, P)** *GAP-Fu* and *cos2 GAP-Fu* clones also lack *smo* activity, and **(N)** *GAP-Fu; Su(fu)* indicates lack of *Su(fu)* activity in the whole disc. Scale bars are 100 µm for **(A, G, I, Q, S)**, 20 µm for **(B–F, H, J, R)**, and 40 µm for all other images. Quantitative results are included in the graphs of Fig 6 for all clones shown and S5 Fig for the AP border of wild-type and *fu^{mH63}* discs.

The result of changing the C-terminal Fu sites to acidic residues in the context of Ci-D1D2 (to form Ci-D1D2D3) was more dramatic. Now, *ptc-lacZ* was restored to control AP border levels (100%) in *fu^{mH63}* mutant discs, albeit with no anterior En induction (Figs 8G, 8H, and S5N). En was, however, induced in *fu^{mH63} Su(fu)* wing discs, contrasting with Ci-WT (S5C and S5D Fig). Also, induction of *ptc-lacZ* was much higher than for Ci-WT or Ci-D1D2 in *cos2* mutant clones (104% of control AP border levels) and En was strongly induced (Figs 6K, 8K, and S5E). These results suggest that Ci-D1D2D3 behaves as, or close to, fully activated Ci in the absence of Fu kinase. It was not, however, constitutively active in anterior cells, where it undergoes apparently normal processing, judged by Ci-155 levels (Fig 8I'). No Ci variant that is processed normally and expressed at physiological levels has been reported to have constitutive activity.

Ci-D1D2D3 activity was induced within anterior clones expressing GAP-Fu, but *ptc-lacZ* expression was lower (19% of control AP border levels) than for Ci-D1D2 (28%) and Ci-WT (30%) (Figs 8L and 6J). The reduction in activity, also apparent in the absence of Su(fu) (Fig 6I and 6L), may be because D3 alterations do not mimic phosphorylation well. The positive effect of Fu on Ci-D1D2D3 in GAP-Fu clones might theoretically be solely due to enhancing Ci-155 levels. On the other hand, S286 and T294 were important for *ptc-lacZ* induction by GAP-Fu in the context of Ci-D1D2, so their phosphorylation would also be expected to contribute to Ci-D1D2D3 activation. If the acidic residues in Ci-D1D2D3 indeed do not mimic phosphorylation very well and there are other activating Fu sites that remain unaltered and unphosphorylated in the absence of Fu kinase activity, it may seem surprising that Ci-D1D2D3 has such strong activity in *fu^{mH63}* wing discs and *cos2* mutant clones. This activity may be deceptive if Fu does not normally achieve close to stoichiometric phosphorylation. Alternatively, it may indicate that Fu can fully activate Ci by using just a subset of available, functionally significant target sites.

## Discussion

Hedgehog signal transduction involves the changing interactions among a limited set of proteins to produce a dose-dependent modulation of Ci/Gli transcription factor activity by (1) reducing the generation of proteolytically processed repressors and (2) increasing the activity of full-length activators [2,13,57,79]. Progress in mechanistic understanding has come principally from a combination of genetic analyses in vivo and studying specific protein interfaces under synthetic conditions. Physiological genetic data regarding the mechanism of activation of full-length Ci/Gli are particularly scarce, partly because the exploration of Ci/Gli variants under normal conditions is challenging and time-consuming in mice, zebrafish, and flies. As demonstrated previously, it is nevertheless essential to investigate Ci/Gli activation under physiological conditions because relative protein stoichiometries are critical; consequently, non-physiological tests often produce misleading results [16,27,51,63,77]. This study provides extensive physiological evidence regarding the mechanism of full-length Ci activation, leading to a new outline model. Key features of the model are that inactive Ci is shielded from interactions required for nuclear transport and transcriptional activator function by Ci–Ci interactions as well as cooperative binding to Su(fu) through at least three interfaces, while phosphorylation by Fu kinase at multiple Ci sites leads to activation by principally opposing repressive Ci–Ci interactions and only limited dissociation of Su(fu). It will be important to test our genetic inferences by identifying specific Ci–Ci interfaces directly within Ci-Su(fu) complexes of various activity states. We nevertheless illustrate our deductions and speculations about Ci activation mechanisms in the discussions below with reference to a plausible specific model (Fig 9).

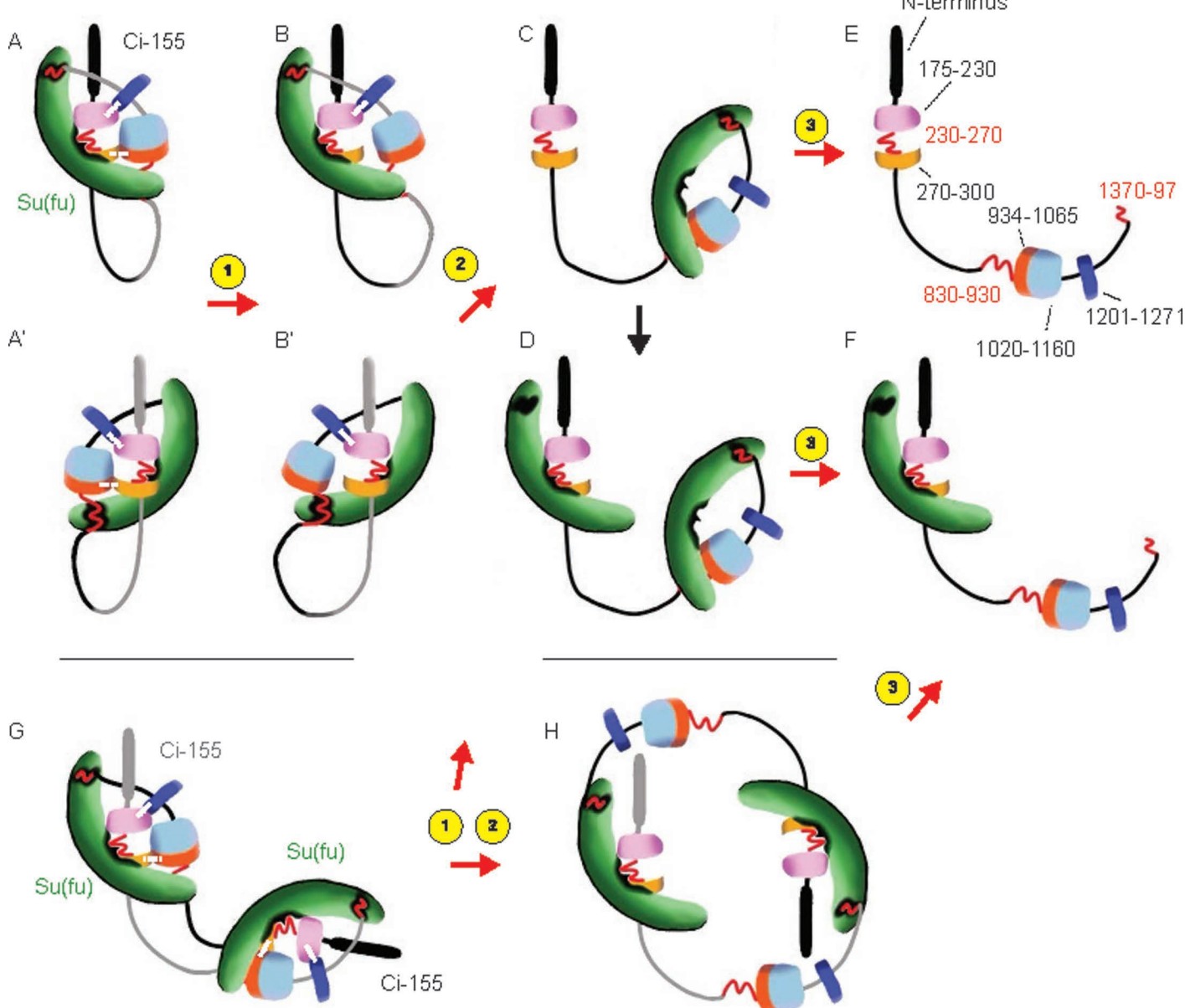

**Fig 9. Potential trajectories for Ci activation by Fused kinase.** The cartoon illustrates a specific speculative rendition of our general model that full-length Ci-155 is held in an inactive state that masks interaction sites for nuclear transport and co-activator binding through a combination of Ci–Ci and Ci–Su(fu) interactions. The regions of Ci involved in Ci–Ci interactions are proposed to surround functionally important Fu target sites around S218 (pink; 175–230), S286/T294 (light orange; 270–300), and S1230 (dark blue; 1,201–1,271). Speculated interactions are indicated by white dotted lines in **A** and **A'** (**A'** and **B'** show views of **A** and **B** structures from the opposite side). Red wavy lines indicate the known Su(fu) binding sites at residues 230–270 and 1370–1397, together with a third binding site that has been mapped to 832–1020, but is shown as 830–930, preceding the Cos2 CORD binding region (934–1065; dark orange) and CBP binding region (1020–1160; light blue). **(A, A')** show potential inert structures if Ci–Ci interactions are intramolecular, stabilizing a structure that allows cooperative binding to Su(fu) (green) at three surfaces. Step 1 shows loss of one Ci–Ci interaction as a consequence of phosphorylation of Fu target sites at S286/T294 and potentially also at another (unidentified or untested) site around the CORD domain to form a structure **(B)** that remains largely inaccessible to nuclear transport or co-activator proteins. The first step could alternatively disrupt the other speculated Ci–Ci interaction by phosphorylation of S218 and S1230 Fu sites (in all cases, followed by CK1 phosphorylation), again without significant loss of Su(fu) affinity or gain of access to other factors (consistent with the properties of Ci-D1D2). A second step of disrupting the second Ci–Ci interaction through further Ci phosphorylation is hypothesized to disrupt the inert Ci conformation that favors cooperative binding to Su(fu). Here, it is speculated that the Ci conformation **(C)** still allows significant cooperative binding at the two weaker Su(fu) binding surfaces. The stronger SYGHI binding site (230–270) may then often bind to another Su(fu) molecule, as in **(D)**. Phosphorylation of S1382 and S1385 may then disrupt local Ci–Su(fu) binding in **(C)** or **(D)** to form structures

(E, F) that no longer favor any cooperative binding of Su(fu) and therefore largely lack associated Su(fu) at the weaker binding surfaces. It is hypothesized that these molecules are no longer protected by Su(fu) from proteolysis, which perhaps depends critically on exposure of C-terminal regions of Ci. (G) shows the potential structure of inert complexes if Ci–Ci interactions are intermolecular, within a dimer. The release of Ci–Ci interactions through Fu phosphorylation to form more accessible structures through steps 1 and 2 could take different paths. Most likely, the Ci–Ci interaction helps a Su(fu) molecule bound to the two weaker sites on one Ci molecule to bind also to the SYGHI region of the second Ci molecule. Loss of these Ci–Ci interactions would then lead to dimer dissociation to form structures shown in (C), possibly followed by binding additional Su(fu) to form (D). Conceivably, however, disrupting Ci–Ci interactions may primarily reduce cooperation between the two weaker Su(fu) binding sites, permitting some formation of a more open dimer structure, as in (H). Further disruption of the interface between Ci 1,370−97 and Su(fu) through additional phosphorylation in step 3 would dissociate this dimeric complex. It is speculated that complexes C, D, or H would have considerable activity and retain substantial protection from proteolysis through Su(fu) binding. Since current evidence suggests that Ci activation per se does not greatly de-stabilize Ci (with reduced *ci* transcription largely responsible for lower Ci-155 levels in En-expressing cells), it is likely that proceeding to structures (E) and (F) with extensive Su(fu) dissociation is rare. If there is no systematically organized ordering of Fu phosphorylation events, then early phosphorylation of S1382 might potentially lead to greater Ci turnover without effective activation.

## Proposed Ci-Su(fu) complexes and activity states overview

Earlier models suggested that Su(fu) binds at multiple sites to keep Ci inactive and protect Ci from degradation. The Ci–Su(fu) protein interfaces include the SYGHI domain (residues 230–272), the extreme C-terminus (residues 1,370–1,397), and at least one more region (currently mapped to 620–1,020 by co-immune precipitation [70] and 832–1,187 by yeast two hybrid [80]; overlap 832–1,020). Our physiological tests suggest that binding to Su(fu) is cooperative; genetic disruption of each of two direct Ci–Su(fu) interfaces significantly increased the activity and decreased the levels of unprocessed Ci. The SYGHI interaction site was also found to limit Ci activity when Su(fu) was covalently linked to Ci, indicating a direct repressive role of that Ci–Su(fu) interface. A single tethered Su(fu) molecule sufficed for inhibition of each Ci–WT-Sufu fusion protein, suggesting that the inactive form of Ci in the absence of Hh is a Ci-Su(fu) complex in 1:1 ratio, with Su(fu) bound at each possible Ci interface. The complexes could involve single Ci and Su(fu) molecules (Fig 9A) or could, for example, be head-to-tail dimers with each Su(fu) molecule interacting with different regions of the two Ci molecules (Fig 9G). These inactive complexes restrict access to other proteins important for Ci to activate transcription, previously suggested to be specifically a nuclear transport protein and CBP co-activator [70]. In this closed complex, there is also limited access to an unidentified degron, which can lead to significant Ci proteolysis in the complete absence of Su(fu).

Here, we make the novel suggestion that Ci conformation, involving specific but presently undefined Ci–Ci interactions, makes an important, regulated contribution to silencing the closed Ci-Su(fu) complex (Fig 9), based on the following evidence. First, deletion of certain residues (270–300, but perhaps also within 175–230) uncoupled Ci activation and loss of protection by Su(fu), suggesting a change in Ci conformation to favor access to proteins required for activation, without substantially altering Ci–Su(fu) interfaces and exposing a Su(fu)-protected degron. Second, Fused kinase can activate Ci in the absence of Su(fu). Third, key phosphorylation sites for Ci activation lie in regions that are either far from any known Su(fu) binding site (S1230) or are in regions (around S218 and S286/T294, flanking the SYGHI binding site) proposed to be involved principally in Ci–Ci interactions. Fourth, the effects of altering those sites on activation by Fu kinase are equally evident in the absence of Su(fu). Fifth, current evidence suggests that Hh signaling in wing discs is not accompanied by a large increase in Ci degradation [60], suggesting only limited changes in protective Ci–Su(fu) interfaces. The model of Ci activation substantially by modulation of Ci–Ci interactions rather than Ci–Su(fu) interfaces differs significantly from earlier hypotheses [48].

There will, nevertheless, likely be interplay between Ci–Ci and Ci–Su(fu) interactions, and one Ci–Su(fu) interface appears to be modulated directly by Fu phosphorylation. Binding of Su(fu) simultaneously to multiple Ci regions likely stabilizes some Ci–Ci contacts and vice versa (Fig 9A and 9G). Thus, disruption of a given Ci–Ci interaction may directly expose Ci to binding an activator (for nuclear transport or a transcriptional co-activator) and may also reduce Ci–Su(fu) binding cooperativity, favoring detachment of Su(fu) from at least one interface (Fig 9C and 9H). A second Su(fu) molecule may, perhaps, now bind at a vacated binding site to keep all Ci–Su(fu) interfaces occupied in a more open Ci

conformation. For example, one Su(fu) molecule may bind the highest affinity SYGHI region, while another still binds cooperatively to the two other binding sites on Ci (Fig 9D). Bound Su(fu) molecules may directly obstruct access to activators to some degree in these partially activated conformations and more completely in the fully closed conformation. Subsequent release of Su(fu) from the C-terminal Ci binding site through local Fu phosphorylation (Fig 9E and 9F) likely reduces direct obstruction of activators by Su(fu) and also de-stabilizes any remaining Ci–Ci interactions, leading to a more open active Ci conformation. Potentially, only the most open Ci conformation with relatively low occupation of Su(fu) binding surfaces (Fig 9E and 9F) is susceptible to degradation, so that only a small fraction of Ci molecules is de-stabilized even at the highest physiological levels of Hh signaling.

Our findings complement the results of a recent study of how Fu kinase accesses Ci [81]. Phosphorylation of Ci within a Ci-Su(fu) complex was found to require a remarkably elaborate set of steps, including Fu autophosphorylation, SUMOylation, and multimerization into condensates that include Ci-Su(fu) and subsequently release phosphorylated Ci to accumulate in the nucleus in complex with Su(fu). Those findings emphasize the robust shielding of Ci phosphorylation sites from Fu kinase in repressive Ci-Su(fu) complexes, consistent with those regions of Ci being involved in Ci–Ci or Ci–Su(fu) interfaces, and add to the evidence that the majority of activated Ci remains associated with Su(fu), consistent with activation principally involving a change in the nature of Ci-Su(fu) complexes. Moreover, the results suggest a rationale for the use of multiple Fu sites on Ci, whereby phosphorylation of each site facilitates access to further sites, resulting in progressively more open conformations.

## Roles of different Fu phosphorylation sites

Individual substitution variants provide an outline for how different Fu sites may cooperate and how they affect Ci activity. Here, we first analyzed S218 and S1230 (simultaneously altering potentially primed surrounding CK1 sites). Ala substituents at both sites virtually eliminated *ptc-lacZ* induction by GAP-Fu, while acidic substituents (Ci-D1D2) did not confer significantly enhanced activity in the absence of Fu kinase activity (*fu^mH63* AP border and *cos2* clones). Ala substituents in the S1230 region had a larger impact than those around S218 at the AP border but the reduced response of each to GAP-Fu appeared to be additive in Ci-A1A2. Phosphorylation of these sites is therefore necessary for a normal response to Fu but insufficient to confer significant Ci activation (with the proviso that acidic residues may not mimic phosphorylation adequately). Ci-D1D2 also did not show any consistent evidence of reduced Ci-155 levels, suggesting no major change in protection by Su(fu) from degradation.

Fu induced *ptc-lacZ* within GAP-Fu clones in wing discs lacking Su(fu) for Ci-WT and Ci-D1D2 but barely detectably for Ci-A1A2 (just as in the presence of Su(fu)). These results show that Fu kinase can activate Ci even in the absence of Su(fu) and that this activation involves both S218 and S1230 phosphorylation and additional residues available for phosphorylation in Ci-D1D2. Su(fu)-independent activation by Fu was previously shown to occur in the absence of proteolytic processing [43]. Here, the elevation of Ci-155 due to GAP-Fu appeared to be at least as high for Ci-A1A2 as for Ci-WT and Ci-D1D2, again suggesting that activity differences cannot be attributed to differences in Ci-155 processing or levels. That leaves two plausible, potentially overlapping mechanisms for activation in the absence of Su(fu): (1) countering inhibition by Cos2 or (2) simply altering Ci to a more active conformation.

In a complementary test, GAP-Fu strongly activated Ci-WT and Ci-D1D2 in *cos2* mutant clones but reduced *ptc-lacZ* induction by Ci-A1A2. Thus, in the absence of both proteolytic processing and Cos2, Ci activation relies on phosphorylation of S218 and S1230, as well as additional Fu sites in Ci-D1D2. The two plausible, potentially overlapping mechanisms for these results are (1) countering inhibition by Su(fu) and (2) altering Ci to a more active conformation. Given these two sets of results (GAP-Fu responses in the absence of Su(fu) or Cos2) and the cited potential mechanisms, the most parsimonious explanation is that S218 and S1230 phosphorylation favor a more active Ci conformation without necessarily directly altering associations with Cos2 or Su(fu). This explanation is also consistent with the locations of S218 and S1230, outside known Su(fu) (230–272, 621–1,020, and 1,370–1,397) and Cos2 ("CDN": 346–440, zinc fingers: 440–620,

"CORD": 940–1,065) binding domains [25,82,83] and suggests that the regions around S218 and S1230 contribute instead to Ci–Ci interactions. It is even possible that the partially additive and partially synergistic effects of S218A and S1230A alterations reflect a direct interaction between these two regions of Ci that is altered by phosphorylation (Fig 9). Other studies have reported only interactions between overexpressed segments of Ci, involving the first 212 residues and the first 2 zinc fingers [84]. The proposed conformational change mediated by phosphorylation of S218- and S1230-containing regions would be a pre-requisite for additional phosphorylation events to turn Ci into an activator sufficient to induce robust *ptc-lacZ* expression.

Alteration of S286 and T294 to Ala and Val, respectively, blocked the response of Ci-D1D2 to GAP-Fu (also in the absence of Cos2 and in the absence of Su(fu)). This might be explained by these residues serving a similar role to S218 and S1230, favoring a Ci conformation compatible with activation. These residues lie within the region (270−300) found to influence Ci activity substantially without any reduction of Ci-155 levels, suggesting largely intact Ci-Su(fu) complexes, consistent with the possibility that their phosphorylation might directly affect Ci–Ci interactions (Fig 9).

The final site investigated, S1382, lies within a small Su(fu) binding region (1,370–1,397). The expectation that phosphorylation might affect Su(fu) binding was previously confirmed using acidic ("D3") substituents in co-immune precipitation of tagged protein segments expressed in tissue culture [52]. Ci-D1D2D3 produced lower levels of Ci-155 than Ci-D1D2 in wild-type and $fu^{mH63}$ wing discs, albeit not quite as low as for Ci Δ1370–1,397, suggesting that the altered Ci–Su(fu) interface is sufficient to expose Ci-155 to increased degradation. While this could reflect significant disruption of all Ci–Su(fu) interfaces because of cooperative binding, it is possible that the Ci-Su(fu) complex is unraveling predominantly around the C-terminus of Ci in response to S1382 phosphorylation and that this region of Ci-155 is critical for Su(fu)-regulated proteolysis. The finding that proteolytically processed Ci/Gli proteins, lacking the C-terminus of full-length proteins, do not appear to bind Su(fu) significantly in vivo or change stability in response to Su(fu) loss, is consistent with this hypothesis. That organization might also allow adequate time for newly synthesized Ci to initiate complex formation with either Cos2 or Su(fu) before proteolytic surveillance for free Ci molecules.

Alteration of S1382 to Ala (Ci-A3) did not reduce the response to Fu in GAP-Fu clones or at the AP border, while addition to C-A1A2 (to form Ci-A1A2A3) did further reduce *ptc-lacZ* induction at the AP border. Thus, it appears that Ci may normally be activated only in part and semi-redundantly by reducing the interface of its C-terminus with Su(fu). This change likely cooperates with alterations to S218, S286, T294, and S1230 to expose Ci in a suitable conformation for interaction with nuclear transport proteins and the co-activator CBP. Thus, the majority of key Fu targets appear to affect Ci–Ci interactions rather than a Ci–Su(fu) interface.

### Ordered phosphorylation events and partial redundancy of the contribution of multiple Fu sites to Ci activation

Although there was no evidence for ordered phosphorylation using uncomplexed Ci as a Fu substrate [52], the discovery that Fu access is highly constrained within Ci-Su(fu) complexes [81] suggests that there could be a relatively ordered set of transitions, allowing access to progressively more Fu sites as the Ci-Su(fu) complex is loosened. We speculate specifically that a set of initial phosphorylation events directly alter Ci–Ci interactions (steps 1 and 2 in Fig 9), leading to a reduction in cooperative Su(fu) binding, so that the C-terminal region of Ci is now more accessible to Fu. Delayed phosphorylation of the C-terminus may then convert Ci into a fully active form, but simultaneously expose it to enhanced degradation (step 3 in Fig 9). We favor this scenario for two reasons. First, earlier, or more extensive, phosphorylation of the C-terminus of Ci relative to other sites might be less effective, leading to a greater de-stabilization of Ci relative to activation than is normally observed at the AP border. This possibility is illustrated by the greater activation of Ci Δ270–300 than Ci Δ1370–97 by GAP-Fu. Second, the properties of Ci variants with acidic substitutions inform the consideration of potentially ordered phosphorylation events. Acidic substituents affect every molecule immediately after synthesis, potentially therefore yielding a stronger output than achieved by natural phosphorylation, which may be far from stoichiometric. This potential is counter-balanced by the likelihood that acidic residues do not mimic phosphorylated residues perfectly

and also by the possibility that ordered phosphorylation normally occurs and is important for efficient Ci activation. The net effect for acidic substitution of S218 and S1230 (in Ci-D1D2) is neutral in GAP-Fu assays, but negative, relative to Ci-WT, in the absence of Su(fu). One interpretation is that acidic residues are a relatively poor mimic at these sites for altering Ci–Ci interactions, but escape from Su(fu) inhibition is unharmed by early phosphorylation of these residues. The response of Ci-D1D2D3 to GAP-Fu (in the presence of Su(fu)) was lower than for Ci-D1D2. This might simply be because D3 is not a good mimic but the result is also consistent with the hypothesis that delayed phosphorylation of the C-terminus normally promotes efficient Ci activation.

One might also ask if all of the relevant Fu sites are included among the four sets tested in this work (acknowledging that each likely nucleates further phosphorylation by CK1 and therefore controls the phosphorylation status of four different regions). On the one hand, Ci-D1D2D3 has strikingly strong activity in the absence of Fu kinase in $fu^{mH63}$ mutant wing discs and in $cos2$ mutant clones, essentially indistinguishable from the effects of removing Su(fu). Together with the evidence from GAP-Fu responses, discussed above, that the net effect of these acidic residues is to mimic phosphorylation imperfectly, this suggests that Ci phosphorylated at these three clusters would be very active. Nevertheless, Ci lacking those phosphorylatable residues (Ci-A1A2A3) was still activated to a significant degree by Fu kinase activity at the AP border. This suggests that other Fu sites can suffice to confer significant activation. Indeed, the contribution of S286/T294 is evident from the failure of GAP-Fu to activate Ci-D1D2av and the presence of at least one additional (unidentified) site is evident from the robust, Fu-kinase-dependent activation of Ci-D1D2avA3 at the AP border. There is therefore likely redundancy, with certain subsets of Fu sites sufficient for strong activation. This is apparent also from the apparently normal activities at the AP border of Ci-A1, Ci-A3, and Ci-D1D2av. The deduction of partial redundancy is also consistent with the speculation that Ci molecules that are not phosphorylated at the C-terminus and remain protected from proteolysis by Su(fu) provide the bulk of Ci activity.

## Contributions of Cos2 to Ci inhibition and activation

Ci-155 must enter the nucleus to activate transcription and interaction with Cos2 likely retains Ci-155 in the cytoplasm [16,85,86]. It is therefore commonly accepted that Cos2 contributes to limiting Ci-155 activity and that this might potentially be altered by Fu kinase activity. Our evidence suggests that the role of Cos2 in Ci activation at the AP border seems likely to be more complex, including a positive component (beyond its requirement for Fu activation by Hh). Every Ci variant that we found to be barely activated by GAP-Fu in anterior cells was much more robustly activated by Fu kinase at the AP border. Moreover, the minimal activation by GAP-Fu was not enhanced in these cases by removing Cos2, as might be expected if the response to Fu at the AP border were simply due to either a greater supply of full-length protein or releasing Ci inhibition by Cos2; in several cases, $ptc$-$lacZ$ induction in $cos2$ mutant clones was actually reduced by GAP-Fu. These observations raise the possibility that Cos2 might facilitate the activation of compromised Ci molecules in GAP-Fu clones and, especially, at the AP border, presumably through direct interactions with Ci.

Cos2 might initially simply compete with Su(fu) for binding to Ci and may bind preferentially to a more open Ci conformation. Under normal conditions (with wild-type Ci), this mechanism may facilitate access of activated Fu (which is initially activated while bound to Cos2) to residues contributing to Ci–Ci interactions in the closed conformation. This would have to be followed by some release of Ci from Cos2 to achieve net steady-state Ci activation. At the AP border, and to a lesser extent in GAP-Fu clones, Ci will not be processed prior to release from Cos2 and there may be increased dissociation of Ci from Cos2 either because of Fu-dependent changes (including Cos2 phosphorylation; see next section) or, specifically at the AP border, through altered interactions of Cos2 with activated Smo that do not involve Fu kinase activity [30,34]. Since Ci-155 activation at the AP border is very low in the absence of Fu kinase, this proposed facilitation mediated by Cos2 is likely only effective if Fu can still phosphorylate some key Ci sites. Thus, we suggest that phosphorylation of a subset of Fu sites can suffice for considerable Ci activation by cycling through Cos2 complexes rather than pure Ci-Su(fu)

complexes *en route* to activation. That mechanism might normally contribute significantly to Ci activity in cells further from the Hh source, where Fu kinase activation is limited [87].

The recent insights concerning Fu kinase access to Ci sites in Ci-Su(fu)-Fu complexes did not examine the potential participation of Cos2 [81]. However, the incorporation of Ci-Su(fu) complexes into Fu condensates presumably renders Ci even less available for other protein interactions and might explain the reduction of activity of many Ci variants deficient for Fu phosphorylation sites by GAP-Fu in the context of *cos2* mutant clones because GAP-Fu would promote condensate formation without significant release of activated Ci.

### Fu kinase effects on Ci processing

There is no documented situation of *ptc-lacZ* induction in wing discs without clear or inferred inhibition of Ci-155 processing (evident here, e.g., by Ci-D1D2D3 or Ci molecules lacking Su(fu) interaction sites). This requirement has been rationalized as necessary to accumulate sufficient full-length Ci for escape from stoichiometric inhibition by Cos2 [18]. Thus, it appears that the normal rate of proteolytic processing can roughly match the rate of Ci-155 synthesis in anterior wing disc cells, with the majority of steady-state Ci-155 complexed to Cos2 if Su(fu) is not present. In the presence of Su(fu), there is a larger steady-state full-length Ci-155 population due to additional Ci-Su(fu) complexes. Potentially, degradation of Ci-155 that is free from both Cos2 and Su(fu) also contributes to silencing of Ci-155 when proteolytic processing is normal.

The response to GAP-Fu involves a modest elevation of Ci-155 that is likely to reflect reduced processing. Full-inhibition of processing (by altering phosphorylation sites required for processing) greatly increases the response to GAP-Fu [16], showing that activation by Fu is certainly sensitive to the degree of Ci-155 processing. We found that none of the Fused Ci sites investigated eliminated the elevation of Ci-155 by GAP-Fu (in fact, the increase appeared stronger for Ci-A1A2). Altering the two known Fu sites on Cos2 (S572 and S931) partially reproduced a phenotype previously reported for Cos2-S572A using overexpressed proteins, of suppressing the elevation of wild-type Ci-155 in response to activated Fu. However, there was still clear elevation of Ci-155 for Ci-A1A2, showing that there must be Fu phosphorylation sites other than the tested Cos2 sites (and Ci sites) contributing to this response. Mechanistically, it might be speculated that the missing sites collaborate with phosphorylation of Cos2 S572 to reduce Ci–Cos2 interaction. The speculative reduced Ci–Cos2 interaction might also contribute to increased Ci activation. This might contribute to the greater activation of Ci-A1A2 and other variants by GAP-Fu when Cos2 is present rather than absent, with Cos2 binding first promoting a more open Ci conformation in competition with Su(fu) and then allowing Ci release.

It is surprising that there is a Ci processing response to Fu kinase at all because proteolytic processing can be fully inhibited at the AP border without Fu kinase and there are also changes in Ci–Cos2 association at the AP border that are likely independent of Fu kinase activity. Conceivably, this phenomenon is a residue of an evolutionarily more primitive response that required a Fu-like kinase for all responses to activated Smo [20].

## Materials and methods

### CRISPR/Cas9 allele cloning

All the CRISPR cloning done in this project is based on the protocol described in Little and colleagues, 2020, [16] and also uses some of the CRISPR fly lines created in that study. In that protocol, a 5kb *mini-white* gene was first inserted into the first intron of the *ci* gene to create a *ci[w+]* allele within the genotype *wlig4 attp40 [nos-Cas9]/Cyo; ci[w⁺]/ci[w⁺]*, which was used for the generation of each new *ci* allele through homologous recombination that replaced a large segment of *ci* coding region and the mini-white gene-bearing first intron. The donor templates with various *ci* mutations were co-injected with guide RNA 3 (gRNA3) GGGCTTACGCCGGTATTAG and guide RNA 4 (gRNA4) GCTTTGGGTGTAGGAGCGTC. The donor template has a 1.1-kb homology region outside of gRNA3 in the 3′ UTR region, and a 2-kb homology region outside

 

of gRNA4 in the first intron. PAM sites of the donor template were altered from GGG to CCG for gRNA 3 and from CGG to CAG for gRNA4. The donor template was cloned into the BSK-F1 Donor construct, as described in [16], and the gRNA sequences were cloned into a dual U6 expression construct pCFD4. These DNAs were then injected into *wlig4 attp40 [nos-Cas9]/Cyo; ci-[w+]/ci-[w+]* fly embryos by Rainbow Transgenic The resulting adults were crossed to *yw hs-flp; Sp/Cyo; TM2/TM6B; Dp[y+]/Dp[y+]* flies, and the progeny were selected for the absence of eye pigment, followed by balancing and genotyping for the introduced DNA changes. "*Dp[y+]*" is an abbreviation for the genotype $Dp(1;4)102[y^+]sv^{spa-pol}$.

### Donor template cloning

All CRISPR *ci* alleles ("CrCi") created were designed using APE software. All the DNA base substitutions, deletions or additions were created via PCR directed mutagenesis using PfuUltraII Fusion HS DNA polymerase (Agilent Technologies). PCR products were inserted into the Zero Blunt Topo cloning vector (Invitrogen) and then re-introduced into the BSK-F1 Donor construct either via restriction digest or Gibson Assembly (New England Biolabs). The final constructs were sequenced using Genewiz service before being sent out for injection.

### *Drosophila* stocks

All fly stocks were maintained in vials with standard cornmeal/molasses/agar medium with baker's yeast supplement at room temperature. For the clonal assays, late first or early second instar larvae were heat shocked at 37°C for 1 hour and dissected 3–4 days later. The crosses for each type of assay are listed below.

**"Wild-type".** Females of the genotype *yw hs-flp; ptc-lacZ/TM6B, Tb; ci⁹⁴/Dp[y+]* were crossed to *yw hs-flp; Sp/Cyo; crCi-X/Dp[y+]* males, selecting third instar larval progeny lacking *y+* and *Tb* to obtain wing discs with a single constructed *crCi* allele as the only source of Ci.

**Su(fu) mutant background.** Females of the genotype *yw hs-flp; Su(fu)^LP (C765-GAL4) ptc-lacZ/TM6B, Tb; ci⁹⁴/Dp[y+]* were crossed to *yw hs-flp; Sp/Cyo; Su(fu)^LP/TM6B, Tb; crCi-X/Dp[y+]* males, selecting third instar larval progeny lacking *y+* and *Tb* to obtain wing discs with a single constructed *crCi* allele as the only source of Ci in a *Su(fu)* null background. *Su(fu)^LP* is a null allele due to an extensive deletion.

**cos2 clones in a Su(fu) mutant background.** Females of the genotype (lab name: '2b') *yw hs-flp UAS-GFP; FRT42D P[Ci+] tub-Gal80/Cyo; Su(fu)^LP C765-GAL4 ptc-lacZ/TM6B, Tb; ci⁹⁴/Dp[y+]* were crossed to males of the genotype *yw hs-flp; FRT42D cos2²/Cyo; Su(fu)^LP/TM6; crCi-X/Dp[y+]*, selecting third instar larval progeny lacking y+ and *Tb* to obtain wing discs with a single constructed *crCi* allele as the only source of Ci in GFP-marked clones lacking *cos2* activity and *P[Ci+]* with neighboring cells expressing *P[Ci+]* (with or without functional Su(fu)). *cos2²* has been considered a null allele [27].

**cos2 clones.** Females of the genotype (lab name: '2R') *yw hs-flp UAS-GFP; smo² FRT42D P[Smo+] tub-Gal80/Cyo; C765-GAL4 ptc-lacZ/TM6B, Tb; ci⁹⁴/Dp[y+]* were crossed to males of the genotype *yw hs-flp; FRT42D cos2²/Cyo; crCi-X/Dp[y+]*, selecting third instar larval progeny lacking *y+* and *Tb* to obtain wing discs with a single constructed *crCi* allele as the only source of Ci and GFP-marked clones lacking *cos2* activity.

**GAP-Fu clones in the presence or absence of Su(fu).** Females of the genotype ('2R') *yw hs-flp UAS-GFP; smo² FRT42D P[Smo+] tub-Gal80/Cyo; (Su(fu)^LP) C765-GAL4 ptc-lacZ/TM6B, Tb; ci94/Dp[y+]* were crossed to males of the genotype *yw hs-flp; smo² FRT42D UAS-GAP-Fu/Cyo; (Su(fu)^LP/TM6B); crCi-X/Dp[y+]*, selecting third instar larval progeny lacking *y+* and *Tb* to obtain wing discs with a single constructed *crCi* allele as the only source of Ci in GFP-marked clones expressing GAP-Fu and lacking *smo* activity (with or without functional Su(fu)). *smo²* is a strong allele that has behaved as a null allele in our prior assays [16,43,88].

**cos2 clones expressing GAP-Fu.** Females of the genotype ('2R') *yw hs-flp UAS-GFP; smo² FRT42D P[Smo+] tub-Gal80/Cyo; C765-GAL4 ptc-lacZ/TM6B, Tb; ci⁹⁴/Dp[y+]* were crossed to males of the genotype *yw hs-flp; smo² FRT42D cos2² UAS-GAP-Fu/Cyo; crCi-X/Dp[y+]*, selecting third instar larval progeny lacking *y+* and *Tb* to obtain wing discs with a single constructed *crCi* allele as the only source of Ci in GFP-marked clones expressing GAP-Fu and lacking *smo* and *cos* activity.

**GAP-Fu clones with *cos2* replacement.** Females of the genotype ('2R') *yw hs-flp UAS-GFP; smo² FRT42D P[Smo+] tub-Gal80/Cyo; C765-GAL4 ptc-lacZ/TM6B, Tb; ci⁹⁴/Dp[y+]* were crossed to males of the genotype *yw hs-flp; smo² FRT42D cos2² UAS-GAP-Fu/Cyo; gCos2-WT or S572A S931A/ TM6B, Tb; crCi-X/Dp[y+]*, selecting third instar larval progeny lacking y+ and *Tb* to obtain wing discs with a single constructed *crCi* allele as the only source of Ci in GFP-marked clones expressing GAP-Fu and lacking *smo* and *cos* activity.

**Loss of Fu kinase with or without Su(fu).** Females of the genotype *yw hs-flp fu^mH63; FRT42D P[y+] P[Fu+]/Cyo; (Su(fu)^LP) C765-GAL4 ptc-lacZ/TM6B, Tb; ci⁹⁴/Dp[y+]* were crossed to males of the genotype *yw hs-flp; Sp/Cyo; (Su(fu)^LP/ TM6B); crCi-X/Dp[y+]*, selecting male third instar larval progeny lacking y+ and *Tb* to obtain wing discs lacking Fu kinase activity (with or without functional Su(fu)) and a single constructed *crCi* allele as the only source of Ci.

**Ci variant with *Su(fu)* replacement, with or without *cos2* replacement.** Females of the genotype *yw hs-flp; (FRT 42D cos2² gCos2 WT or S572AS931A)/ (FRT 42D cos2² gCos2 WT or S572AS931A); Su(fu)^LP C765-GAL4 ptc-lacZ/ TM6B; ci⁹⁴/Dp[y+]* were crossed to *yw hs-flp; (FRT 42D cos2² gCos2 WT or S572AS931A)/ (FRT 42D cos2² gCos2 WT or S572AS931A); (UAS-Su(fu)5A or UAS-mSufu); crCi-X/Dp[y+]* selecting third instar larval progeny lacking y+ and *Tb* to obtain wing discs with a single constructed *crCi* allele as the only source of Ci, and UAS-Su(fu) expression replacing endogenous *Su(fu)*, with or without a *gCos2* allele replacing endogenous *cos2*. For replacing Su(fu) with UAS-Su(fu)-WT, a second chromosome transgene was used in the cross *yw hs-flp; Su(fu)^LP C765-GAL4 ptc-lacZ / TM6B; ci⁹⁴/Dp[y+]* crossed to *yw; UAS-Su(fu)-WT / P[y+] CyO); Su(fu)^LP; crCi-X/Dp[y+]*.

## Immunohistochemistry

Late third instar larvae were collected and inverted for wing discs. They were fixed in 4% paraformaldehyde in PBS for 30 min and rinsed three times with PBS, blocked with 10% normal goat serum (Jackson ImmunoResearch Laboratories, Inc.) in 0.1% PBS-Triton (PBS-T) at room temperature, and then incubated overnight in primary antibodies solutions with 1% normal goat serum and 0.1% PBS-T at 4°C. For *ptc-lacZ* protein product staining ("Ptc-lacZ"), rabbit anti-ß-galactosidase (MP Biomedicals) in 1:10,000 dilution was used. For Ci-155 staining, Rat 2A1 anti-Ci (Developmental Studies Hybridoma Bank) in 1:3 dilution was used. For En staining, Mouse 4D9 anti-Engrailed (Developmental Studies Hybridoma Bank) in 1:4 dilution was used. Once stained overnight, they were washed three times in PBS-T and then incubated in secondary antibody solutions for an hour to two hours at room temperature. Alexa Fluor 488, 546, 594, or 647 secondary antibodies (Molecular Probes) were used accordingly, in 1:1,000 dilution. Then, they were washed once in PBS-T and washed once in PBS. Afterward, discs were mounted in Aqua/Poly mount (Polysciences).

## Imaging and quantitation

Fluorescence images were captured at 20× and 63× using 1.4 NA oil immersion lenses on LSM 700 and 800 confocal microscopes (Carl Zeiss). For each staining set, the laser intensity was set using control wild-type discs, such that maximum signals at the AP border did not reach saturation at any point (using the range indicator). Those settings were then used for all samples in the set. All experimental AP border and clone Ptc-lacZ and Ci-155 intensity values were expressed as fractions of control disc AP border values from the same experiment, as described in detail below. Image J (NIH, Bethesda, Maryland) was used to quantify the fluorescent intensity for both AP border and anterior clone measurements.

For measurements of Ptc-lacZ (*ptc-lacZ* protein product) and Ci-155 at the AP border, the fluorescence intensity profile was taken from anterior to posterior along a narrow rectangle spanning the wing disc pouch region, avoiding the D/V border (where Notch and Wnt signaling affect Hh signaling output) and any disc deformities. Ptc-lacZ is induced only at the AP border and there is no *ci* expression in the posterior compartment. Background values were subtracted from the peak intensity value at the AP border, using the posterior wing disc values for Ci-155 and the anterior wing disc values for Ptc-lacZ (because there is occasional artifactual posterior Ptc-lacZ) for each disc as background, including the control discs. Then, the corrected Ptc-lacZ and Ci-155 peak values for each experimental disc were divided by the average of

the equivalent values for the control discs in the staining set to give values as a fraction of controls. The mean values for all discs of a genotype relative to controls were then calculated, whether discs were all in one staining set or distributed among different staining sets. SEMs were calculated, where the *N* value is the number of experimental discs.

For AP border measurements, the controls were wild-type discs with one copy of a wild-type *ci* allele, with the exception of experiments conducted for Figs 2 and S4. For Fig 2, *fu*^mH63^; S*u(fu)*/ *Su(fu)* discs with one copy of a wild-type *ci* allele were used as the controls for experimental Ci variant genotypes. A separate experiment measured *fu*^mH63^; S*u(fu)*/ *Su(fu)* AP border values versus wild-type controls (both with one copy of wild-type Ci). Experimental genotype results were multiplied by these "correction" ratios to derive a value normalized to the usual condition of a wild-type control. In S5 Fig, the Ci-WT result for *fu*^mH63^ discs derived from experiments with wild-type discs is used, just as for the genotypes to which it is compared (hence, the small difference in *fu*^mH63^ Ci-WT values between Figs 2 and S5). For S4 Fig, wild-type discs with two copies of Ci-WT were used as the control.

For clonal assays, the average value of the fluorescence intensity was taken from circular regions of multiple anterior clones (on average, five per disc), non-clonal anterior regions (three per disc) for Ptc-lacZ background, and non-clonal posterior regions (three per disc) for Ci-155 background. All measurements were taken from z-sections with maximal AP border signals in the wing pouch. Clones and the circular region sampled were chosen to avoid the D/V border and any folds, with almost all lying in the wing pouch. Fluorescence intensity was also measured within a small rectangular section of the AP border of control discs with two copies of wild-type Ci (measurements at the AP border of experimental discs were also taken and reported in supplementary spreadsheets but not used in the calculation of normalized clone intensities). The width of the rectangular section of the AP border for Ptc-lacZ measurement was equal to the average width of the strongly elevated Ptc-lacZ stripe among control discs. This generally corresponded to the top 15 percent of Ptc-lacZ values along the profile. These rectangles were then positioned so that the center aligns with the maximum Ptc-lacZ value for each disc. The width of the rectangle for Ci-155 measurement with 2A1 antibody was defined by the average width of AP border Ci-155 from the peak value to half the peak value (further anterior) among controls. These rectangles were then positioned so that the posterior end aligns with the maximum Ci-155 value for each disc. The average width of Ci-155 rectangles was roughly twice the width of Ptc-lacZ rectangles (15.6 vs. 7.5 pixels). All clone and AP border intensity values were then corrected for background by subtracting the average anterior value for Ptc-lacZ and average posterior value for Ci-155 of the same discs. The corrected fluorescent intensity value of each clone was then divided by the corrected average AP border values of control discs in the same experiment. The mean ratio for all clones of a given genotype was then calculated, whether discs were all in one staining set or distributed among different staining sets. SEMs were calculated, where the *N* value is the number of clones.

The ratio for the Ci-155 value of anterior cells outside any clones was obtained in a similar fashion (and used in Fig 7), using the anterior Ci-155 value instead of clonal values for each genotype. The average of the anterior Ci-155 values for each disc was obtained and corrected for background, and then, the ratio of the anterior values was derived relative to the control AP border value. The mean of those ratios was calculated, with the *N* value for SEM corresponding to the number of discs of the experimental genotype.

Quantitative results for each Ci variant in each assay were compared with Ci-WT, estimating the significance of differences using Student's two-tailed unpaired *t* test with Welch correction for unequal sample numbers. Selected additional pairs of Ci variants or genetic conditions were collected for comparisons with the same test. All comparison results can be found in supplementary spreadsheets. Since comparisons for a particular condition (e.g., *cos2* clones) included up to 9 Ci variants over the whole study, we denoted comparisons with $p < 0.005$ in graphs, using a black asterisk for comparisons to Ci-WT and a red asterisk for all other comparisons.

## Adult wings

Adult wings were dissected from anesthetized flies and then washed in 70% ethanol followed by 100% ethanol and then mounted in Aqua/Poly Mount (Polysciences). They were imaged on the LSM 700 confocal microscope (Carl Zeiss).

## Statistics and reproducibility

All images presented are representative of at least three examples. The sample size was based on prior experience and previously published protocols. No samples were excluded from analysis unless there was an issue with the quality of the staining. All samples were treated in the same manners without preconception or prejudice of the possible outcomes. The standard error of the mean was used to determine the errors for individual values and significant differences determined and shown as explained in the "Imaging and quantitation" section above.

## Supporting information

**S1 Table. Tabulation of all CRISPR-engineered *ci* alleles used with DNA changes and consequent changes in amino acid sequence.**
(XLSX)

**S1 Fig. (Related to Fig 1). Loss of Su(fu) binding sites or two SYGHI-adjacent regions increase Ci activity in *cos2* mutant clones: effects on En induction.** (**A**) Key Ci features are illustrated. The top cartoon shows Su(fu) binding regions (red), the zinc finger domain (ZF, which binds DNA and can bind Cos2), and the binding region for CBP co-activator. The second cartoon shows deletions employed in this study (pink, yellow, and blue) and the CORD Cos2 binding domain. The third cartoon shows Fu phosphorylation sites examined in this study. PKA sites (S838, S856, S892) that promote Ci-155 processing, and a third Cos2 binding region (CDN; 346–440) are not shown. (**B–H**) Third instar wing discs (20× objective for (**B**) and 63× objective for all other images) with one copy of the indicated *ci* CRISPR allele, GFP marking homozygous *cos2* mutant clones (green; yellow arrowheads), and yellow dotted lines marking the AP border. (**B'–H'**) Ptc-lacZ expression (red) and (**B"–H"**) En protein (gray-scale) in the same discs. In (**B**) the entire wing, disc lacks Su(fu) activity. Scale bars are 100 μm for (**B**) and 40 μm for all other images.
(DOCX)

**S2 Fig. (Related to Fig 4). Su(fu) inhibition is increased by covalent linkage to Ci but still requires non-covalent binding to the SYGHI region of Ci: effects in PKA mutant clones.** (**A–C**) Third instar wing discs (63× objective) with one copy of (**A, C**) Ci-WT-Sufu or (**B**) Ci-SYAAD-Sufu, GFP marking *pka* mutant clones (green; yellow arrowheads), and yellow dotted lines marking the AP border. (**A'–C'**) Ptc-lacZ expression (red) and (**A"–C"**) Ci-155 expression (gray-scale) in the same discs. (**C**) Su(fu) activity was absent in the whole disc. Scale bars are 40 μm. (**D**) Bar graph showing the average ratio of Ptc-lacZ intensity and Ci-155 intensity in clones relative to the AP border of wild-type control discs, together with SEMs (*n* values 94, 30, and 23, respectively, for each set of three genotypes). Differences with $p < 0.005$ (Student $t$ test with Welch correction) are indicated for comparing to Ci-WT-Sufu in an otherwise wild-type disc (red asterisk). (**E**) Bar graph showing the average ratio of Ci-155 intensity in the indicated clones relative to the AP border of wild-type control discs, together with SEMs (*n* values 20, 5, 14, 10, 87, 19, 40, 54, 46, 47, 199, and 90); wing disc images and *ptc-lacZ* graph for these clones are in Fig 4. Differences with $p < 0.005$ (Student $t$ test with Welch correction) are indicated for comparing a Ci variant to Ci-WT (black asterisk) or for comparisons between bracketed pairs (red asterisk). Please see Materials and methods for details of measurements and expression of all experimental values relative to AP border values of control wild-type wing discs. The data underlying the graphs shown in the figure can be found in S4 Data.
(DOCX)

**S3 Fig. (Related to Fig 5). S218 and S1230 each contribute to activation by Fu and GAP-Fu reduces the activity of several Ci deletion variants in *cos2* mutant clones, as for Ci-A1A2.** (**A, B, F–M**) Third instar wing discs (20× objective for (**G**) and 63× objective for other images) with the named Ci variant, GFP marking the indicated clone types (green; yellow arrowheads), and yellow dotted lines marking the AP border. (**C–G**) All have Ci Δ1,201–1,271, while (**A, B**) have *GAP-Fu* clones and (**J-M**) have *cos2 GAP-Fu* clones. (**A', B', F'–M'**) Ptc-lacZ expression (red) and (**A", B", F"–H",**

J"–M") Ci-155 expression (gray-scale) in the same discs. (**A, B, G, I–M**) *GAP-Fu* and *cos2 GAP-Fu* clones also lack *smo* activity. (**A, B**) Both S218A (Ci-A1) and S1230A (Ci-A2) reduced Ptc-lacZ induction by GAP-Fu. (**C–E**) Third instar wing discs with one copy of Ci Δ1,201–1,271, showing Ptc-lacZ (red) and (**C', E'**) Ci-155 expression (gray-scale) (20× objective) or (**D'**) En expression (green) (63× objective), resembling Ci-WT behavior, with the AP border marked by dotted yellow lines. (**F, G**) Induction of Ptc-lacZ and Ci-155 levels for Ci Δ1,201–1,271 also resembled Ci-WT in (**F**) *cos2* and (**G**) *GAP-Fu* clones. (**H–M**) The addition of GAP-Fu in *cos2* mutant clones (**I'**) increased Ptc-lacZ (compare to Fig 1E) and (**I"**) En induction by Ci-WT but decreased Ptc-lacZ induction by (**H', J'**) Ci ΔΔ (which lacks residues 175–230 and 1,201–1,271) and by (**K'–M'**) Ci variants lacking residues 1,370–97, 175–230, or 270–300 (compare to Fig 1H, I, K). Scale bars are 40 μm for (**A, B, F, H–M**), 100 μm for (**C, E, G**), and 20 μm for (**D**).
(DOCX)

**S4 Fig.** (**Related to Fig 7**). **Loss of Fu sites in Su(fu) and Cos2 do not further reduce activity of Ci lacking S218 and S1230 Fu sites.** (**A, D, F, I**) Third instar wing disc with one copy of Ci-A1A2 (*crCi-A1A2/ci⁹⁴*), showing (**A, D, F, I**) Ptc-lacZ (red) and (**A', D', F', I'**) Ci-155 expression (gray-scale) (20× objective). The discs are (**A**) otherwise wild-type or (**D, F, I**) *Su(fu)ᴸᴾ/ᴸᴾ* but expressing the indicated *UAS-Su(fu)* transgene using *C765-GAL4.* (**B, E, G, J, M–O**) Third instar wing disc with two copies of Ci-A1A2, showing (**B, E, G, J, M, O**) Ptc-lacZ (red) and (**B', E', G', J', M', O'**) Ci-155 expression (gray-scale) (20× objective) or (**C, H K, N**) Ptc-lacZ (red) and (**C', H', K', N'**) En expression (green) (63× objective; AP border marked by dotted yellow lines). The discs are (**B, C**) otherwise wild-type, (**E, G, J**) *Su(fu)ᴸᴾ/ᴸᴾ* but expressing the indicated *UAS-Su(fu)* transgene using *C765-GAL4* or (**M–O**) additionally lack endogenous *cos2* activity but contain one copy of the genomic transgene *gCos-AA* (encoding S572A S931A alterations). Su(fu)-5A has Fu site Ser residues substituted by Ala; mSufu encodes mouse Sufu. Scale bars are 20 μm for (**C, H, K, N**) and 100 μm for all other images. (**L**) Bar graph showing the ratio of Ptc-lacZ intensity at the AP border of the named genotypes (with Cos-AA replacement of endogenous Cos2 in pink) relative the AP border of wild-type discs, together with SEMs ($n = 29, 16, 27, 20, 17, 3, 22, 36, 28$, and 15, respectively, for wing discs with Ci-A1A2). The blue dotted line at 1.0 marks the Ci-WT value. Differences with $p < 0.005$ (Student $t$ test with Welch correction) are indicated for comparing *ptc-lacZ* at the AP border for Ci-A1A2 together with the indicated Su(fu) variant, or additional Cos2 variant, to Ci-A1A2 alone (separately for either one copy or two copies of Ci-A1A2) (red asterisk). Please see Materials and methods for details of measurements and expression of all experimental values relative to AP border values of control wild-type wing discs. The data underlying the graphs shown in the figure can be found in S7 Data.
(DOCX)

**S5 Fig.** (**Related to Fig 8**) **Contributions of S286, T294, S1382, and S1385 to Ci activation.** (**A–D, J–L**) One copy of named Ci variants in (**A, B, J, K**) otherwise wild-type, (**C, D**) *fuᵐᴴ⁶³ Su(fu)ᴸᴾ/ᴸᴾ* or (L) *fuᵐᴴ⁶3* third instar wing discs, showing (**A–D, J–L**) Ptc-lacZ (red), (**A', J', L'**) Ci-155 (gray-scale), or (**B'–D', K'**) En expression (green, with the AP border marked by dotted yellow lines). (**E–I**) Third instar wing discs (63× objective) with one copy of (**E**) Ci-D1D2D3, with Ptc-lacZ (red) and En (green) expression shown in *cos2* clones (yellow arrowheads), or with (**F, G**) Ci-A1A2A3 or (**H, I**) Ci-D1D2avA3, GFP marking (**F-I**) *smo GAP-Fu* clones (green; yellow arrowheads) in otherwise (**F, H**) wild-type or (**G, I**) *Su(fu)ᴸᴾ/ᴸᴾ* discs, and yellow dotted lines marking the AP border. Scale bars are 100 μm for (**A, J, L**), 20 μm for (**B, C, D, K**), and 40 μm for all other images. (**M**) Bar graph showing the ratio of Ptc-lacZ intensity at the AP border for the named Ci variants and Su(fu) genotypes relative to the AP border of control discs, together with SEMs ($n = 20, 17, 15, 29, 22, 23$, and 7, respectively). (**N**) Bar graph showing the ratio of Ptc-lacZ intensity at the AP border for the named Ci variants in otherwise wild-type (red) and *fuᵐ⁶³* wing discs (pink) relative to the AP border of control discs, together with SEMs ($n = 29, 15, 24, 39, 13, 17$, and 12, respectively, for wild-type and 38, 5, 7, 21, 7, 20, 14, and 10, respectively, for *fuᵐᴴ⁶³*). The difference between pairs of values shows the contribution of Fu kinase activity at the AP border. (**M** and **N**) Differences with $p < 0.005$ (Student $t$ test with Welch correction) are indicated for comparing a Ci variant to Ci-WT (black asterisk) or comparisons

between bracketed pairs (red asterisk). Please see Materials and methods for details of measurements and expression of all experimental values relative to AP border values of control wild-type wing discs. The data underlying the graphs shown in the figure can be found in S8 Data.
(DOCX)

**S1 Data. Raw and processed data spreadsheets related to Fig 1N and 1O.**
(XLSX)

**S2 Data. Raw and processed data spreadsheets related to Fig 2I and 2J.**
(XLSX)

**S3 Data. Raw and processed data spreadsheets related to Fig 3G and 3H.**
(XLSX)

**S4 Data. Raw and processed data spreadsheets related to Figs 4L, S2D, and S2E.**
(XLSX)

**S5 Data. Raw and processed data spreadsheets related to Fig 6J, 6K, and 6L.**
(XLSX)

**S6 Data. Raw and processed data spreadsheets related to Fig 7G and 7H.**
(XLSX)

**S7 Data. Raw and processed data spreadsheets related to S4L Fig**
(XLSX)

**S8 Data. Raw and processed data spreadsheets related to S5M and S5N Fig**
(XLSX)

## Acknowledgments

We thank Brian Heubel, Jennifer Ding, and Aaron Choi for research assistance and continued discussion and input, Brian Heubel and Amy Reilein for review and editing, the Bloomington Stock Center and Rainbow Transgenic Flies for provision of genetic reagents, the Developmental Studies Hybridoma Bank (DSHB) for antibodies, FlyBase as an information resource, and the confocal microscope resource provided by the Department of Biological Sciences, Columbia University.

## Author contributions

**Conceptualization:** Hoyon Kim, Jamie C. Little, Daniel Kalderon.

**Data curation:** Hoyon Kim, Daniel Kalderon.

**Formal analysis:** Hoyon Kim, Daniel Kalderon.

**Funding acquisition:** Daniel Kalderon.

**Investigation:** Hoyon Kim, Jamie C. Little, Jiashen Li, Bryna Patel, Daniel Kalderon.

**Methodology:** Hoyon Kim, Jamie C. Little, Daniel Kalderon.

**Project administration:** Hoyon Kim, Jamie C. Little, Daniel Kalderon.

**Resources:** Daniel Kalderon.

**Software:** Hoyon Kim.

**Supervision:** Hoyon Kim, Jamie C. Little, Daniel Kalderon.

**Validation:** Hoyon Kim, Daniel Kalderon.

**Visualization:** Hoyon Kim.

**Writing – original draft:** Hoyon Kim, Daniel Kalderon.

**Writing – review & editing:** Hoyon Kim, Daniel Kalderon.

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
