## [Editor Report · Decision Letter 0]

29 Aug 2024

Dear Dr Kalderon, 

Thank you for submitting your manuscript entitled "Physiological analysis of the mechanism of Ci transcription factor activation through multiple Fused phosphorylation sites in Hedgehog signal transduction" for consideration as a Research Article by PLOS Biology. Please accept my sincere apologies for the delay in getting back to you with feedback as we consulted with an academic editor about your submission. 

Your manuscript has now been evaluated by the PLOS Biology editorial staff, as well as by an academic editor with relevant expertise, and I am writing to let you know that we would like to send your submission out for external peer review.

Once your full submission is complete, your paper will undergo a series of checks in preparation for peer review. After your manuscript has passed the checks it will be sent out for review. To provide the metadata for your submission, please Login to Editorial Manager (https://www.editorialmanager.com/pbiology) within two working days, i.e. by Aug 31 2024 11:59PM.

Kind regards,

Richard

Richard Hodge, PhD

rhodge@plos.org

PLOS

---

## [Decision Letter · Decision Letter 1]

25 Nov 2024

Dear Dr Kalderon,

Thank you for your patience while your manuscript "Physiological analysis of the mechanism of Ci transcription factor activation through multiple Fused phosphorylation sites in Hedgehog signal transduction" was peer-reviewed at PLOS Biology. Please accept my sincere apologies for the long delays that you have experienced during the peer review process. Your manuscript has now been evaluated by the PLOS Biology editors, an Academic Editor with relevant expertise, and by an independent reviewer. Please note that the Academic Editor handling your manuscript has provided a review and this is labelled as 'Comments from the Academic Editor' for full transparency. We had recruited an additional reviewer to provide a review here, but unfortunately this report was not submitted so we have decided to proceed at this stage to avoid any additional loss of time. 

In light of the reviews, which you will find at the end of this email, we would like to invite you to revise the work to thoroughly address the reviewers' reports.

As you will see, Reviewer #1 and the Academic Editor are positive about your study and think it provides valuable insights into endogenous Hh signal transduction. They both ask that additional quantifications, clarifications and reporting details are included. In addition, Reviewer #1 raises concerns with the overall strength of the biochemical validation of the Ci-Ci interaction model and requests that Ptc-luciferase assays are added. After discussions with the Academic Editor, we will not make these experiments essential for the revision and we ask that you please caveat this limitation and acknowledge that the model is based on inferences from mutational analysis, highlighting the need for future validation using biochemical and/or structural methods. However, we would encourage you to provide some biochemical evidence to strengthen the model if the experiments are not too difficult to perform. Finally, the Academic Editor notes that the discussion section should be streamlined to make it easier to follow for non-experts. Further to this, we would also like to take this opportunity to ask that you please ensure that the manuscript text is generally written with a broad readership in mind, due to the complex nature of the pathway studied. 

Given the extent of revision needed, we cannot make a decision about publication until we have seen the revised manuscript and your response to the reviewers' comments. Your revised manuscript is likely to be sent for further evaluation by all or a subset of the reviewers.

**IMPORTANT - SUBMITTING YOUR REVISION**

*Re-submission Checklist*

*Published Peer Review*

*PLOS Data Policy*

*Blot and Gel Data Policy*

Sincerely,

Richard

Richard Hodge, PhD

rhodge@plos.org

REVIEWS:

Reviewer #1 (Jianhang Jia, signs review): 

Comments

The manuscript by Kim et al. describes complex models for Ci regulation by Sufu, Fu, and Cos2 through analyzing the individual and combined effects of Fu phosphorylation residues in Ci. The authors developed an impressive in vivo model to dissect the activity of various forms of Ci in inducing Ptc-lacZ expression in Drosophila wing disc. One model suggest that Ci-Ci interaction is regulated through Ci phosphorylation by Fu, although additional supporting materials are needed. By mutating a series of Fu phosphorylation sites in endogenous Ci, the authors sophisticatedly analyzed Ci activity under various conditions such as Fu mutation, Sufu mutation, and/or Cos2 mutation. The study provides extensive data and physiological relevance for the mechanisms of Ci regulation by multiple factors including Fu, Sufu, and Cos2, potentially offering a comprehensive understanding of Ci regulation in vivo. However, the manuscript contains several misleading descriptions that need to be carefully addressed. Additional experiments are needed to address Ci-Ci interaction, while other concerns can be addressed by rewriting the statements and discussion. 

Major concerns:

This manuscript needs editing for typos and confusing statements (some examples are listed below).

Ci-Ci interaction model: The authors proposed an intriguing model for Ci-Ci interaction therefore activation independent of Sufu regulation; however, there is no further evidence to support this idea. Does Ci interact with itself in co-immune precipitation, co-localization, and/or in vitro interaction assays? Does Hh signaling regulate Ci-Ci interaction in this model? Additionally, does Ci-Ci interaction apply to Ci155-Ci75?

Figure 1F-F", needs larger clones. The changes in Ci level might not be visible in small clones.

Figure 1M&N, need statistical analysis to indicate the significance. SEM is insufficient to identify the significance among samples. This concern applies to all similar figures. 

For all the figures, the authors should label the staining with colored text to make it easier to follow. 

Figure 3, if the authors want to use Figure 3G-H to compare the levels of Ptc-lacZ and Ci155, they need to show the density data of Ptc-lacZ and Ci155 at the border. 

A Ptc-luciferase assay would be more sensitive compared to the wing disc. The authors would have used some cultured cells for such an assay.

How did the authors compare Ptc-lacZ and Ci expression across various wing discs? They used two Zeiss confocal microscopes, possibly using the same software and conditions for image acquisition. How were imaging conditions normalized? Were laser intensity and other parameters normalized?

Line 642-643 mentions 77% and 66%. How can readers judge these numbers if they are not shown in the figure? Although some information can be dig out from supplemental files, it is not easily comprehensible. This issue applies to all such figures. 

Regarding Ci phosphorylation by Fu, although the dose-dependent model is well accepted, the data presented in this study does not exclude the possibility of sequential phosphorylation. Does combining CiA sites with CiD sites change Ci activity? These experiments can be tedious because of the many sites to consider; however, the authors should at least discuss this possibility.

Minor concerns:

Figure 1E-E'', explain why some of the cos2 clones exhibit higher levels of Ci staining by the 2A1 antibody.

Line 192, what about Fu null mutant or using a dominant-negative Fu? 

Figure 1 D-D', needs a higher magnification to show Ci levels near the border. 

Figure1, for all figures indicating Ci variants, the authors may want to use crCi or crCixxx to indicate that this form of Ci is not an overexpression, similar to the name "gCos2". 

Line 273-274, the statement "…suggesting that the higher Ci levels in the first three variants report greater Ci stability" is confusing, based on the data shown in Figure 1M and Figure2J.

Figure 2A", the image was out of focus. 

Figure 2E-E', this disc is significantly larger. Is that because of the deletion of 175-230? Explain. 

Line 294-298, confusing statement.

Line 300, "interface" or "interaction"?

Line 311-312, what "clones"? How did the authors measure Smo activity in "anterior clones"? 

Line 317-320, the statement "The increase in Ci-155…..by GAP-Fu" is a confusing. 

Figure 3, how were the 42% and 30% calculated? 

The ci allele with Ci-Sufu fusion is a smart design. Is Ci processing changed by fusing Sufu to the C-tail of Ci. Although their lab might not use cultured cells, Ci processing can also be detected using wing disc samples.

Do the authors have a control for Ci-Sufu, e.g., adding another sequence to Ci C-tail? 

How does the 1:1 Ci:Sufu ration hypothesis apply to anterior cells far away from the AP border?

Figure 4, Ci-WT-Sufu equals to Ci-SuFu, correct?

Line 478, Ci-delta-delta has a slightly greater activity deficit than Ci-A1A2. Does this mean the deleted amino acids have additional positive roles in regulating Ci? How to further confirm this?

The use "SuFu" needs to be consistent throughout the manuscript. The authors can possibly use "dSufu" for fly and "mSufu" for mouse. 

Line 568 "Fu-EE, which has acidic residues…" should be described when Fu-EE first appears. 

Line 597, define the name "gCos-AA"

Figure 7, Needs to show Ptc-lacZ to mark the border. 

Figure 8A', Ci is very bright with normal staining pattern. How the authors standardize the procedure for taking images?

Figure 8 K-O", remove the yellow lines on the top. Those are not the focus.

Figure S1B, what about cos2 clones away from the AP border? Also in S1B legend, "second line shows deletions…..binding domain" needs clarification.

COMMENTS FROM THE ACADEMIC EDITOR

The relationship between Fu kinase, Su(Fu), Cos2, and Ci transcription factors during Hedgehog (Hh) signal transduction has remained unresolved. This question has been largely investigated in cultured cells using overexpressed proteins - led to numerous propositions, but to what degree they apply to endogenous Hh signaling under physiological conditions is unclear. In this study, Kim et al. utilize a Drosophila wing imaginal disc model, a well-established system for Hh signaling, to address this gap by generating CRISPR-based mutations in endogenous pathway components, thereby avoiding some of the pitfalls of prior overexpression strategies. This approach allowed them to evaluate various mutations in Ci, which are implicated in Fu phosphorylation, Su(Fu) binding, or both.

Principal Findings

-Multiple regions on Ci, including areas around previously defined Su(Fu) binding sites, contribute to Ci activation.

-Fu's activation of Ci does not strictly require Su(Fu) binding or Cos2 interaction. This supports the idea that Fu phosphorylation enables Ci to adopt an activated conformation, rather than merely escaping Su(Fu) inhibition.

-Fu phosphorylation of Cos2 and Su(Fu) is largely dispensable for Ci activation.

-The identified Fu phosphorylation sites do not fully explain Fu's action on Ci, implying the existence of additional phosphorylation sites that have not yet been mapped.

Whilst some of these findings have been reported earlier, the present study adds value by validating them using endogenous pathway components in a physiological system.

Recommendation: Support publication in PLOS Biology pending the following revisions:

-Quantification of Figures: Quantification is missing from several key figures. For example, the deletions of residues 301-320 and 321-346 in Fig. 1 are not quantified, and the text refers to deletions of residues 175-232, while the figure panel mentions 175-230. Additionally, figures 5 and 8 lack any quantification. These omissions should be addressed to ensure the results are properly substantiated.

-Streamlining the Discussion: The Discussion is overly long and difficult to follow, particularly for non-experts. I recommend simplifying it by adding a concise summary paragraph near the beginning that highlights the principal findings and their conceptual significance. I also suggest trimming redundant sections for clarity.

-Structural Model in Figure 9: While the data generally support the model that Ci undergoes a conformational change, the structural model presented in Fig. 9 is difficult to follow and reads as highly speculative, because it is largely based on inferences from mutational analysis rather than direct biophysical or structural data, and no three-dimensional structural model of Ci is available. These limitations should be more clearly acknowledged in the Discussion.

-Clarification of GAP-Fu: The manuscript refers to GAP-Fu, which, according to the referenced literature, is a membrane-targeted Fu construct. This should be clearly defined within the text to avoid confusion for readers.

With these relatively minor revisions, I believe the manuscript will significantly contribute to the field by clarifying the endogenous roles of Fu, Su(Fu), and Cos2 in Hh signal transduction.

---

## [Editor Report · Decision Letter 2]

3 Mar 2025

Dear Dr Kalderon,

Thank you for your patience while we considered your revised manuscript "Physiological analysis of the mechanism of Ci transcription factor activation through multiple Fused phosphorylation sites in Hedgehog signal transduction" for publication as a Research Article at PLOS Biology. This revised version of your manuscript has been evaluated by the PLOS Biology editors and the Academic Editor.

Based on our Academic Editor's assessment of your revision, I am pleased to say that we are likely to accept this manuscript for publication, provided you address the following editorial requests that I have provided below (A-B). Thank you for already providing the source data underlying the figures as a supplementary file, this means that our requests are not particularly extensive. 

(A) We routinely suggest changes to titles to ensure maximum accessibility for a broad, non-specialist readership. In this case, we would suggest the following edit to the title, as follows. I appreciate that the manuscript reports several findings that provide evidence for a new model of Ci activation, but we feel this is the most striking finding that could be highlighted in the title? Please ensure you change both the manuscript file and the online submission system, as they need to match for final acceptance:

“Ci is activated by alterations in Ci-Ci interfaces without full Suppressor of Fused dissociation during Hedgehog signaling”

(B) Per journal policy, if you have generated any custom code during the course of this investigation, please make it available without restrictions. Please ensure that the code is sufficiently well documented and reusable, and that your Data Statement in the Editorial Manager submission system accurately describes where your code can be found.

We expect to receive your revised manuscript within two weeks. 

*Published Peer Review History*

*Press*

Best regards,

Richard

Richard Hodge, PhD

rhodge@plos.org

PLOS

---

## [Editor Report · Decision Letter 3]

7 Mar 2025

Dear Dr Kalderon,

On behalf of my colleagues and the Academic Editor, Benjamin Myers, I am pleased to say that we can accept your manuscript for publication, provided you address any remaining formatting and reporting issues. These will be detailed in an email you should receive within 2-3 business days from our colleagues in the journal operations team; no action is required from you until then. Please note that we will not be able to formally accept your manuscript and schedule it for publication until you have completed any requested changes.

Thank you for providing a more fleshed out version of our suggested title, I agree that this provides some additional important information and we are happy to change it. For the data files, the Excel files are appropriately named although typically the figure legends include a sentence at the end such as the following:

'The underlying data can be found in Figure 2_data' 

During the production process, I would be grateful if you could update the figure legends to include this, which removes the 'supplementary information' part of the sentence. I did not think it was worth holding the acceptance of your paper for a minor formatting edit like this, so I have gone ahead and passed your manuscript over to production. 

PRESS

Best wishes, 

Richard

Richard Hodge, PhD

rhodge@plos.org

PLOS
